# Cellular senescence in malignant cells promotes tumor progression in mouse and patient Glioblastoma

Rana Salam [1,8], Alexa Saliou[1,8], Franck Bielle [1,2,3], Mathilde Bertrand[4], Christophe Antoniewski[5], Catherine Carpentier[1], Agusti Alentorn [1,6], Laurent Capelle[7], Marc Sanson[1,3,6], Emmanuelle Huillard[1], Léa Bellenger[5], Justine Guégan [4] & Isabelle Le Roux [1] ✉

Glioblastoma (GBM) is the most common primary malignant brain tumor in adults, yet it remains refractory to systemic therapy. Elimination of senescent cells has emerged as a promising new treatment approach against cancer. Here, we investigated the contribution of senescent cells to GBM progression. Senescent cells are identified in patient and mouse GBMs. Partial removal of p16[Ink4a]-expressing malignant senescent cells, which make up less than 7 % of the tumor, modifies the tumor ecosystem and improves the survival of GBM-bearing female mice. By combining single cell and bulk RNA sequencing, immunohistochemistry and genetic knockdowns, we identify the NRF2 transcription factor as a determinant of the senescent phenotype. Remarkably, our mouse senescent transcriptional signature and underlying mechanisms of senescence are conserved in patient GBMs, in whom higher senescence scores correlate with shorter survival times. These findings suggest that senolytic drug therapy may be a beneficial adjuvant therapy for patients with GBM.

Diffuse gliomas are the most common primary malignant brain tumors in adults[1]. Glioblastoma (GBM; IDH-wild type glioma, grade 4) is the most aggressive glioma and despite intensive conventional therapy which includes surgery, radiation, and both concurrent and adjuvant temozolomide (TMZ) chemotherapy, GBM remains treatment-resistant and disease progression is fatal, with a median survival below 15 months[2]. Distinct factors may account for current treatment failures, including tumor invasiveness, an immunosuppressive microenvironment and intra-tumoral heterogeneity. Novel approaches are therefore required to find effective therapeutic strategies.

Cellular senescence is a permanent cell cycle arrest mediated by p53/p21[CIP1] and/or p16[INK4A]/Rb pathways and is defined by a combination of features including a senescence-associated secretory phenotype (SASP), anti-apoptotic program, and increased lysosomal content, the latter allowing histochemical detection of senescence associated-β-galactosidase activity (SA-β-gal)[3]. In cancer, cellular senescence is triggered by multiple stresses such as DNA damage, oncogene activation, therapeutic agents, or elevated reactive oxygen species (ROS). SASP is defined by the secretion of a plethora of factors including cytokines, chemokines, growth factors, extracellular matrix (ECM) components, and proteases, which together can stimulate

[1]Paris Brain Institute (ICM), Hôpital Pitié-Salpêtrière, Inserm U 1127, CNRS UMR 7225, Sorbonne Université, Genetics and Development of Brain Tumors Team, Paris, France. [2]AP-HP, Hôpital de la Pitié-Salpêtrière—Charles Foix, Département de Neuropathologie, Paris, France. [3]Paris Brain Institute (ICM), Hôpital Pitié-Salpêtrière, Inserm U 1127, CNRS UMR 7225, Sorbonne Université, Onconeurotek Tumor Bank, Paris, France. [4]Paris Brain Institute (ICM), Hôpital Pitié-Salpêtrière, Inserm U 1127, CNRS UMR 7225, Sorbonne Université, Data Analysis Core Platform, Paris, France. [5]Sorbonne Université, CNRS FR3631, Inserm US037, Institut de Biologie Paris Seine (IBPS), ARTbio Bioinformatics Analysis Facility, Paris, Institut Français de Bioinformatique (IFB), Paris, France. [6]AP-HP, Hôpital de la Pitié-Salpêtrière—Charles Foix, Service de Neurologie 2-Mazarin, Paris, France. [7]AP-HP, Hôpital de la Pitié-Salpêtrière—Charles Foix, Service de Neurochirurgie, Paris, France. [8]These authors contributed equally: Rana Salam, Alexa Saliou. ✉e-mail: isabelle.leroux@icm-institute.org

angiogenesis, modulate the composition of the ECM and promote an epithelial-to-mesenchymal transition[4,5]. Depending on the context, senescence exerts two opposite effects during tumorigenesis[6]. In some contexts, senescent cells prevent the proliferation of pre-malignant cancer cells, as SASP factors stimulate the immune clearance of onco-gene or therapy-induced senescent tumor cells[7–9]. Conversely, in per-sistently senescent cells, the SASP can either directly induce tumor growth[10] or contribute to immune suppression, thus allowing tumor progression[11,12]. Many studies have assessed the function of senescence in developing tissues and age-related diseases by the in vivo removal of senescent cells, either using chemical or genetic senolytics[13–17]. A common genetic approach employs *p16^Ink4a* regulatory sequences to drive the inducible expression of INK-ATTAC or p16-3MR, which selectively eliminate senescent cells expressing *p16^Ink4a*, leading to apoptosis[16,17]. This senolytic strategy efficiently reduces the adverse effects of therapy-induced senescent cells including cancer recurrence in a mouse breast cancer model[18].

A few in vivo studies have begun to examine the role of cellular senescence in gliomas. Using mouse patient-derived xenograft (PDX) models, it was shown that IL6, a universal SASP component as well as a cytokine expressed by immune cells, promotes the growth of patient glioma stem cells (GSCs) and contributes to glioma malignancy[19]. Conversely, loss of PTEN-PRMT5 signaling induces senescent GSCs to slow down tumorigenesis[20]. Furthermore, loss of one allele of p53 reduces H-RasV12 oncogene-induced senescence in an orthotopic GBM model, as evidenced by reduced SA-β-gal staining, and decreases mouse survival time[21]. Finally, a recent study revealed the dual effect of therapy-induced senescence (TIS) following BMI1 inhibitor treatment of diffuse intrinsic pontine glioma tumor (a pediatric high-grade glioma), which initially attenuates tumor cell self-renewal and growth, but later leads to SASP-mediated tumor recurrence[22]. This study con-firmed the detrimental function of persistent senescent cells in glial tumors and suggested that senescent cells could represent an actionable target to mitigate the process of gliomagenesis[6].

Recent single-cell RNA sequencing studies classified the intra-tumoral heterogeneity of malignant GBM cells[23–27], which can be sub-divided into four main cellular plastic states: oligodendrocyte pro-genitor cell-like (OPC-like), neural progenitor cell-like (NPC-like), astrocyte-like (AC-like) and mesenchymal-like (MES-like) states[23]. The relative abundance of these cellular states within the tumor defines three GBM transcriptomic subtypes, with proneural (PN-GBM) and classical (CL-GBM) GBMs associated with neurodevelopmental pro-grams and mesenchymal GBM (MES-GBM) associated with injury response programs[24–29]. OPC-like and NPC-like states are enriched in PN-GBM whereas AC-like and MES-like states are enriched in CL-GBM and MES-GBM, respectively[23]. Notably, stemness programs are het-erogeneous even within a single tumor, and PN- and MES-GSCs could contribute to the genetic heterogeneity observed in patient GBM[24,26,30]. Each transcriptomic GBM subtype is associated with distinct molecular alterations and patient outcomes. MES-GBM is correlated with enhanced activation of anti-inflammatory (or tumor-promoting) macrophages[29,31–33]. Mutations in *NF1*, *TP53*, and *PTEN* genes, and increased NF-κB signaling are prevalent in this GBM subtype[28]. Inter-estingly, *PTEN* loss induces cellular senescence and activates NF-κB signaling, which initiates and maintains the SASP[34–36]. Together these findings support the idea that cellular senescence could contribute to the intra-tumoral heterogeneity of GBM.

In this study, we investigated whether cellular senescence parti-cipates in GBM tumor progression using patient-resected GBM tissues and a mouse GBM model[37]. We identify senescent cells in patient and mouse GBMs. Partial removal of senescent cells expressing high levels of p16^Ink4a using a ganciclovir-inducible p16-3MR transgenic line[17] improves the survival of GBM-bearing mice. To identify the cells expressing high levels of p16^Ink4a, and to characterize the action of these cells on the tumor ecosystem, we combined single-cell and bulk

RNA sequencing (RNAseq) analysis at early and late time points after the senolytic treatment. This approach leads to the identification of the NRF2 transcription factor and its selected targets as a signal trig-gering the pro-tumoral activity of p16^Ink4a expressing senescent cells. Using these data, we define an unbiased senescence signature that we successfully used to interrogate GBM patient data sets, revealing that higher senescence scores correlate with shorter survival times.

## Results

### Identification of senescent cells in patient and mouse gliomas
We first searched for senescent cells in 28 freshly resected diffuse gliomas from patients by performing SA-β-gal staining coupled with immunohistochemistry (IHC) (14 GBMs, 5 astrocytomas, 9 oligodendrogliomas; Supplementary Fig. 1a). Ninety-eight percent of SA-β-gal positive (SA-β-gal+) cells were negative for the cell cycle marker Ki67 (Ki67−) and 72% of SA-β-gal+ cells were positive for the cell cycle inhibitor p16^INK4A, strongly suggesting that the majority of SA-β-gal+ cells were senescent (Fig. 1a; SA-β-gal+ Ki67−: $98.35 \pm 1.96\%$; SA-β-gal+ p16^INK4A+: $72.51 \pm 10.52\%$; $n = 4$ p16^INK4A-non deleted tumors). In gliomas harboring a mutation of p53, few SA-β-gal+ cells expressed high levels of mutant p53 (Fig. 1a and Supplementary 1a) showing that some SA-β-gal+ cells were malignant cells. To identify further senescent cells, we used cell type-specific markers. Some SA-β-gal+ cells co-expressed GFAP, which could either represent parenchymal astrocytes or malignant cells, OLIG2, an OPC marker, or IBA1, a microglia/macrophage marker (Fig. 1a). To establish a quantitative measure of senescent cell burden, we quantified the percentage of the tumor area containing SA-β-gal+ cells, and used these measures to stratify tumors into three senescent categories: (1) >1% ($n = 10/28$) but below 7%; (2) ≤1–0.1%> ($n = 13/28$); and (3) ≤0.1% ($n = 5/28$) senescent cells (Supplementary Fig. 1a and b). Notably, no diffuse glioma types nor the sex of the patient were associated with a particular senescent category.

We next investigated whether specific molecular alterations were associated with each of the senescent tumor categories, as defined by SA-β-Gal cell percentages. Homozygous deletion of *CDKN2A*, encoding for p16^INK4A, is carried by 54% of patient GBMs (cbioportal.org). As p16^INK4A is a mediator of senescence, we annotated p16^INK4a status of each tumor, as well as examined other common molecular alterations, including p53, PTEN, NF1, and EGFR mutations. Notably, we did not find any association of a specific molecular alteration with a single senescent category (Supplementary Fig. 1a).

We then studied senescence in an immuno-competent GBM mouse model, employing a modified version of a model developed by Friedmann-Morvinski et al.[37]. This model recapitulates the molecular alterations identified in MES-GBM: the loss of *Pten* and *p53* and the inactivation of *Nf1* triggered by the ectopic expression of H-RasV12 (Fig. 1b). Six-to-eight week-old Glast^creERT2/+;Pten^fl/fl female mice were intracranially injected with a lentivirus encoding H-RasV12-IRES-eGFP and shp53, into the subventricular zone (SVZ). Mice were sacrificed when they reached disease endpoints, hereafter referred to as late timepoint (Fig. 1b). These tumors displayed a heterogenous histo-pathology similar to that described in patient GBMs[38] (Supplementary Fig. 1c). By qPCR analysis, elevated expression levels of *Ink4/ARF* (encoding p16^Ink4a, p15^Ink4b, p19^Arf) and *p21*, both of which encode senescence-mediating proteins, were detected in the tumor (GFP+) cells compared with the surrounding parenchyma (GFP−) cells (Fig. 1c). In contrast, *p53* mRNA levels were similarly low within GFP+ tumor cells and adjacent GFP− cells, in agreement with the presence of shp53 in the lentivirus (Fig. 1c). We observed, similarly to patient tumors, that 95% of SA-β-gal+ cells were Ki67− ($94.56 \pm 2.27\%$, $n = 6$) and 75% were p19^ARF+ ($75.17 \pm 4.37\%$; $n = 5$). Again, these results strongly suggest that the majority of SA-β-gal+ cells were senescent. Notably, p16^Ink4a protein could not be examined in mouse tissues due to the lack of a suitable

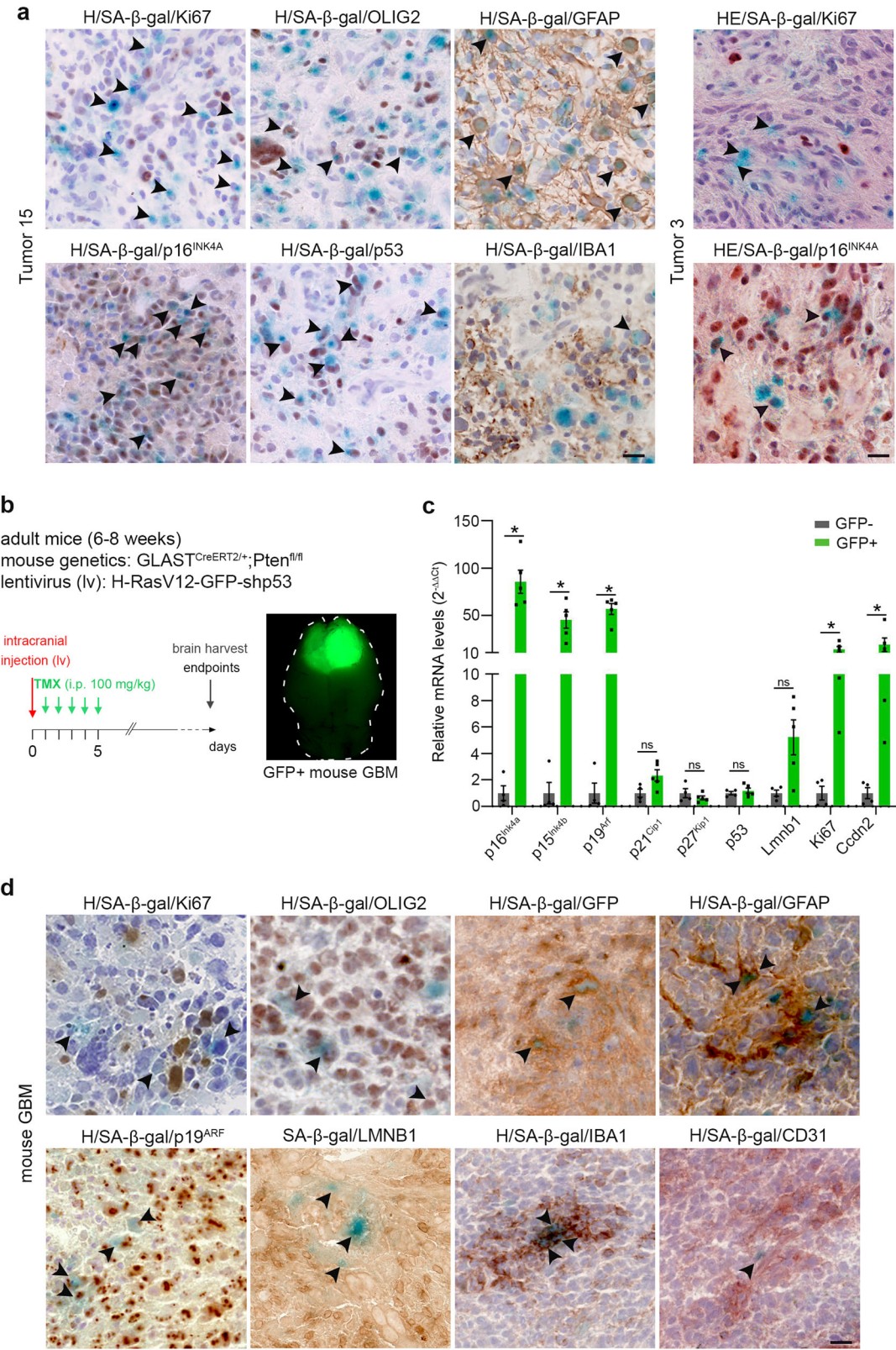

antibody. We also identified SA-β-gal+ LAMINB1− cells, illustrating the characteristic loss of the nuclear envelope integrity of senescent cells. Within the murine tumor, senescent cells were of distinct types, and included either malignant (GFP+) tumor cells, glial cells (GFAP+, OLIG2+), or microglia/macrophage (IBA1+) (Fig. 1d). We did not detect any senescent endothelial cells (CD31+; Fig. 1d). In general, the senescent cells were sparsely distributed in the tumor, and mostly located in

proliferative areas or adjacent to necrotic regions (Supplementary Fig. 1d).

Altogether these data reveal that cellular senescence is associated with primary gliomagenesis, including in the mouse GBM model, which recapitulates the histopathology, senescence features, and cell identities of patient GBMs. We thus further used this model to address the function of senescent cells during primary gliomagenesis.

**Fig. 1 | Identification of senescent cells in patient and mouse gliomas.**
**a** Representative SA-β-gal staining (blue) coupled with immunohistochemistry (IHC, brown) on two non-fixed patients GBM cryosections samples (Ki67 and GFAP: $n = 16$; p16[INK4A]: $n = 12$; IBA1: $n = 10$; p53: $n = 7$ and OLIG2: $n = 6$ patient GBMs). **b** Left: genetics of the mouse mesenchymal GBM model (mouse and injected lentivirus (lv)). The timeline represents the induction of tumorigenesis with tamoxifen intraperitoneal (i.p.) injections (TMX, 100 mg/kg/day for 5 days). Brains are harvested when mice reach endpoints. Right: the representative stereomicroscopic image of a mouse brain with a GFP+ GBM. **c** Relative transcript levels shown as ratios of normalized values of mouse GBM (GFP+, $n = 5$) over surrounding

parenchyma (GFP−, $n = 4$). Data are represented as the mean ± SEM. Statistical significance was determined by the Wilcoxon–Mann–Whitney test (*$p < 0.05$; ns, not significant). **d** Representative SA-β-gal staining (blue) coupled with IHC (brown) on mouse GBM cryosections. (Ki67, p19, IBA1 and GFP: $n = 8$; GFAP: $n = 6$; OLIG2 and CD31: $n = 5$; LMNB1: $n = 4$ independent mouse GBMs). Arrowheads in **a**, **d** point to SA-β-gal+ cells co-labeled for the markers OLIG2, GFP, GFAP, p19[ARF], or IBA1. For Ki67, LMNB1, and CD31 IHC, the arrowheads point to mono-labeled SA-β-gal+ cells. Scale bars, **a** and **d**: 20 μm. H hematoxylin, HE hematoxylin, and eosin, i.p. intraperitoneal, TMX tamoxifen. Raw data are provided as a Source Data file.

## Senescent cell's partial removal increases the survival of GBM-bearing mice

We introduced the *p16-3MR* transgene in the mouse GBM model, which allowed the selective removal of senescent cells expressing *p16[Ink4a]* with ganciclovir (GCV) injections[17]. Levels of the *p16-3MR* transgene were elevated in the tumor (GFP+) cells compared with the surrounding parenchyma (GFP−) cells, similar to *p16[Ink4a]* expression, suggesting that in our model *p16-3MR* expression followed the same trend as the endogenous expression of *p16[Ink4a]* (Supplementary Fig. 2a). Remarkably, the median survival of GBM-bearing mice harboring *p16-3MR* that were treated with GCV (p16-3MR+GCV) increased significantly compared with WT mice treated with GCV (WT+GCV) or p16-3MR mice treated with vehicle (p16-3MR+vhc) (Fig. 2a–c). Similarly, the survival of GBM-bearing mice treated with the senolytic drug ABT263 (Navitoclax, an inhibitor of the anti-apoptotic proteins BCL2 and BCL-xL[13]) increased significantly compared with control mice (WT+vhc) (Fig. 2b and d). Together these results strongly suggest that senescent cells act as a pro-tumoral mechanism during primary gliomagenesis.

To confirm the tumor-promoting function of senescent cells, we further studied GBM mice carrying the p16-3MR transgene. First, we analyzed whether senescence hallmarks decreased in p16-3MR+GCV GBMs compared with controls at the late time point (i.e., disease endpoint). We quantified the percentage of the tumor area (defined by GFP expression) encompassing SA-β-gal cells and found that it decreased 2.2-fold (from 2.26% to 1.02%) in p16-3MR+GCV tumors compared with WT+GCV GBMs (Fig. 2e and f). On average, about 2% of the tumor area was comprised of SA-β-gal cells in WT+GCV GBMs, which corresponds to senescent category one as we defined using patient gliomas (Supplementary Fig. 1a).

We next performed bulk RNA sequencing (RNAseq) of the tumors with or without senescent cells. In agreement with the inter-tumoral heterogeneity of patient GBMs, heat maps of the bulk RNAseq data revealed inter-tumoral heterogeneity of mouse GBMs independent of the treatment (Supplementary Fig. 2b and d). Gene set enrichment analysis (GSEA, Supplementary Fig. 2f) of p16-3MR+GCV GBMs compared with WT+GCV GBMs revealed an upregulation of cell cycle components (E2F targets), a downregulation of pathways involved in cancer (Notch signaling, mTORC1 signaling, epithelial–mesenchymal transition, angiogenesis), and modulation of the immune system (TNFA signaling via NFKB, Interferon responses, Il2-Stat5 signaling). In addition, bulk RNAseq analysis revealed a slight decrease in *p16[Ink4a]* levels in p16-3MR+GCV GBMs compared with control GBMs (Fig. 2g). By qPCR analysis, significantly decreased expression levels of *p16[Ink4a]*, *p15[Ink4b]*, *p19[Arf]* were detected in treated GBMs compared with control GBMs (Fig. 2h). Finally, GSEA revealed a significant downregulation of senescence pathways (Fig. 2i, Supplementary Fig. 2g; Supplementary Data 1). SASP genes whose expression was significantly decreased in p16-3MR+GCV compared with WT+GCV GBMs included *Fn1*, *Plau*, *Timp1*, *Ereg*, and *Bmp2*[39–41], the qPCR analysis further validated *Ereg* decrease (Fig. 2j, Supplementary Fig. 2c, e, and h). These SASP genes encode growth factors and extracellular matrix components or remodelers.

Collectively our data show that at the late timepoint, when mice were sacrificed due to tumor burden, there was an increased survival of GBM-bearing mice associated with the partial removal of p16[Ink4a] senescent cells, therefore pointing to the tumor-promoting action of senescent cells during gliomagenesis.

## Identification of p16[Ink4a Hi] cells in a subset of malignant cells

To unveil the identity of senescent cells expressing *p16[Ink4a]* and targeted by the p16-3MR transgene with GCV[17], we performed droplet-based single cell RNAseq (scRNAseq) on FACs sorted cells from WT and p16-3MR GBM cells collected 7 days after the last GCV injection, hereafter named early timepoint (Fig. 3a and b; Supplementary Data 1). At this stage, WT+GCV GBMs ($n = 2$) exhibit increased tumor growth compared with p16-3MR+GCV GBMs ($n = 2$) (Supplementary Fig. 3a–c). Uniform manifold approximation and projection (UMAP) clustering at 0.5 resolution revealed 22 clusters with distinct gene expression signatures in each sample in the two conditions (Fig. 3c and Supplementary Fig. 3d). Non-malignant cells were identified based on the expression of the pan-leukocyte marker *CD45* (*Ptprc*), and malignant cells were identified by their expression of the 3′ long terminal repeat (3′*LTR*) of the injected lentivirus and by copy number variations (CNV) (Fig. 3d and Supplementary Fig. 3e). Cells in each of the 22 clusters expressed variable levels of *p16[Ink4a]* (*Cdkn2a*) however, only malignant tumor cells expressed high levels of *p16[Ink4a]*. The p16-3MR mice were used in different cellular contexts. Injection of GCV always decreased significantly *p16[Ink4a]* levels. However, this decrease never exceeded *p16[Ink4a]* basal expression levels corresponding to the levels observed in the organ in absence of senescent cells[17,18]. We hypothesized that cells expressing *p16[Ink4a]* at a level ≥4 (hereafter, we refer to p16[Ink4a Hi] cells) were the cells targeted by p16-3MR with GCV (Fig. 3d). This threshold was chosen as p16[Ink4a Hi] cells represent 3% (412/13563) of the tumor cells, a percentage that is in agreement with the area of SA-β-gal staining in the tumors (Supplementary Fig. 1a and Fig. 2f).

We then focused our analyses on the malignant cell compartment. The p16[Ink4a Hi] cells were mostly present in cluster 0 which comprises the highest cell number in WT+GCV GBMs (2910 out of 13 563 cells; Fig. 3d). Further UMAP clustering of malignant cells at 0.6 resolution identified 17 clusters in the two conditions (Fig. 3e). GSEA using the mouse gene lists published by Weng et al.[42] allowed the malignant cell clusters to be assigned cellular identities, which predominantly included cycling cells, pri-oligodendrocyte progenitor cell-like (pri-OPC-like), committed OPC-like (COP-like), myelinating oligodendrocyte (mOL), astrocyte-like (AC), neural progenitor-like (NP-like), and hypoxic cells (HC) (Fig. 3e and f, Supplementary Fig. 3f; Supplementary Data 1). The labeling of the clusters was also in agreement with GSEA using human gene lists published by Bhaduri et al.[26] (Supplementary Fig. 3g). Some clusters exhibited mixed cell identities. The astrocyte cluster shared gene signatures of astrocytes, endothelial cells, and ependymal cells whereas the pri-OPC-like 1 (pOPC1) and pri-OPC-like 2 (pOPC2) clusters shared gene signatures of pri-OPC-like cells, astrocytes and COP cells (Fig. 3f). Of note, the enrichment score of each subpopulation differed very little between p16-3MR+GCV and WT+GCV GBMs, except for the pOPC1-3 clusters (Fig. 3f).

The p16[Ink4a Hi] cells were mainly grouped in the astrocyte cluster and to a lesser extent in the NP-like cluster (Fig. 3g). The levels of *p16[Ink4a]* decreased significantly in the astrocyte cluster in p16-3MR

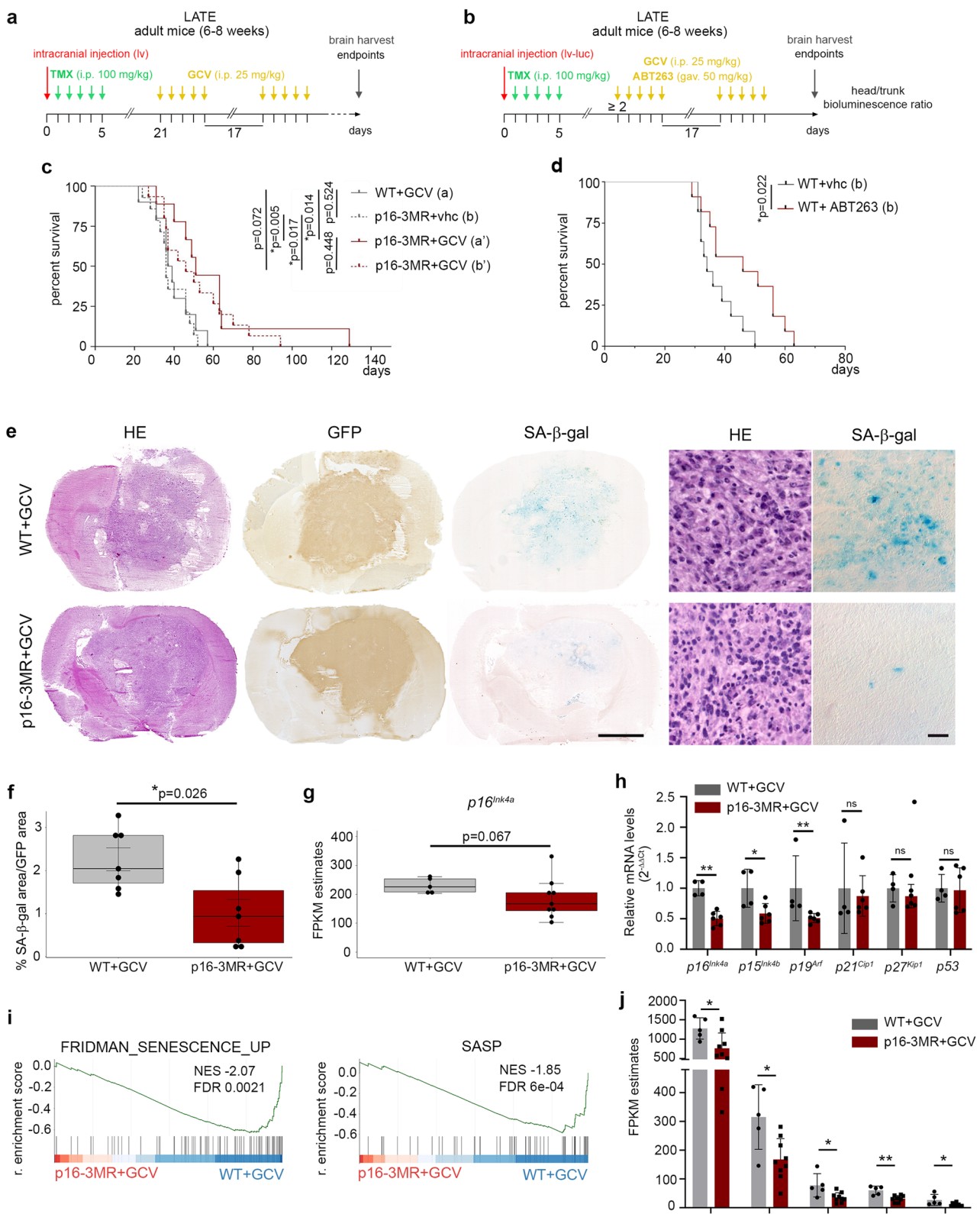

+GCV GBMs compared with WT+GCV GBMs. No other malignant and microenvironment (CD45+) clusters showed a significant difference in *p16^Ink4a* levels between the two conditions (Fig. 3h; Supplementary Data 2). Therefore, this analysis identifies the astrocyte cluster as senescent. On line with a senescent phenotype, the astrocyte cluster shared an inflammatory signature (gene signatures of microglia and tumor-associated macrophages) (Supplementary Fig. 3g). Remarkably,

the percentage of cells in the astrocyte cluster decreased in p16-3MR +GCV GBMs compared with WT+GCV GBMs (astrocyte cluster from 7.75% to 3.21%; Fig. 3i), in agreement with the partial removal of p16^Ink4a Hi cells by the p16-3MR transgene in the presence of GCV.

Altogether, scRNAseq analysis identifies senescent malignant p16^Ink4a Hi cells belonging to the astrocyte cluster displaying inflammatory phenotype and targeted by the p16-3MR with GCV.

**Fig. 2 | Senescent cell's partial removal increases the survival of GBM-bearing mice. a** Timeline of tumorigenesis induction (lv: H-RasV12-GFP-shp53) and removal of senescent cells with GCV, 21 days post lv injection in the p16-3MR transgenic mouse. **b** Timeline of tumorigenesis induction (lv-luc: H-RasV12-GFP-P2A-Luc2-shp53) and removal of senescent cells with GCV in the p16-3MR mouse or with ABT263 in WT mouse when head to body bioluminescence ratio reached 2. **c** Kaplan–Meier survival curves (solid lines) of WT ($n = 10$, median survival 38 days) and p16-3MR ($n = 9$, median survival 51 days) mice treated with GCV as shown in (**a**). Kaplan–Meier survival curves (dotted lines) of p16-3MR mice treated with vhc ($n = 14$, median survival 36 days) or GCV ($n = 15$, median survival 46 days) as shown in (**b**). **d** Kaplan–Meier survival curves of WT mice treated with vhc ($n = 11$, median survival 34 days) or ABT263 ($n = 11$, median survival 46 days) as shown in (**b**). **e** Representative HE, GFP IHC, and SA-β-Gal staining on adjacent mouse GBM cryosections. Right panels represent higher magnifications of the left panels. Scale bars, left panels: 2.5 mm, right panels: 20 μm. **f** Quantification of the SA-β-gal area over the tumor (GFP+) area ($n = 7$ biologically independent animals/group). **g** Relative transcript levels of *p16Ink4a*, shown as FPKM estimates extracted from bulk

RNAseq analysis (WT+GCV, $n = 5$; p16-3MR+GCV, $n = 9$). **h** Relative transcript levels are shown as ratios of normalized values of p16-3MR+GCV GBMs ($n = 6$) over WT+GCV GBMs ($n = 4$). **i** GSEA graphs from bulk RNAseq data in p16-3MR+GCV GBMs compared with WT+GCV GBMs. SASP gene list from Gorgoulis et al.[3] (Supplementary Data 1). **j** Relative transcript levels of genes in WT+GCV and p16-3MR+GCV GBMs extracted from bulk RNAseq data (WT+GCV, $n = 5$; p16-3MR+GCV, $n = 9$). **c** and **d** Statistical significance was determined by Mantel–Cox log-rank test. **b** Raw *p*-values from the log-rank tests are included in the figure and significance is indicated by * for *p*-values below the 5% level after correction by the Benjamini–Hochberg procedure. **f–h, j** Data are represented as the mean ± SD and statistical significance was determined by the Wilcoxon–Mann–Whitney test ($*p < 0.05$; $**p < 0.01$, ns, not significant). TMX tamoxifen, vhc vehicle, gav. gavage, GCV ganciclovir, i.p. intraperitoneal, lv lentivirus, lv-luc lentivirus-luciferase, GSEA gene set enrichment analysis, FDR false discovery rate, NES normalized enrichment score, r. enrichment score running enrichment score. Raw data are provided as a Source Data file.

## Partial removal of p16^Ink4a Hi malignant cells impacts the remaining malignant cells

We next analyzed whether the partial removal of p16^Ink4a Hi cells impacted the remaining malignant cells in our GBM model. Although the size of the tumors at this early timepoint differed between p16-3MR +GCV and WT+GCV GBMs (Supplementary Fig. 3a–c), the percentage of cycling cells (pOPC3, G2/M1, G2/M2, G1/S1, G2/S2 clusters) remained stable (WT+GCV: 28.78%; p16-3MR+GCV: 28.05%). In addition, three clusters of the oligodendroglial lineage pOPC2, COP, and mOL, increased in cell proportions upon the partial removal of p16^Ink4a Hi cells (pOPC2 from 6.81% to 13.10%; COP from 1.24% to 4.60%; mOL from 1.31% to 4.30%; Fig. 3i). We further validated these results using bulk RNAseq data of WT+GCV and p16-3MR+GCV GBMs collected at the early timepoint (Supplementary Fig. 3h). GSEA using the Weng et al.[42] gene lists showed no difference in the expression of cycling genes (Supplementary Fig. 3i). In contrast, there was an increase of transcripts associated with COP and mOL gene signatures upon the partial removal of p16^Ink4a Hi cells (Fig. 3j). The increase in cell numbers (scRNAseq) and in gene signatures (bulk RNAseq) of the oligodendroglial lineage suggests a shift of the malignant cellular states upon senolytic treatment. Indeed, GSEA revealed an increase in OPC-like and NPC-like states and their associated proneural transcriptional subtype following the partial removal of p16^Ink4a Hi cells. In parallel, GSEA showed a decrease in the MES-like state and the mesenchymal transcriptional subtype (Fig. 3j). Of note, no significant change in transcripts associated with the stemness gene signature defined by Tirosh et al.[43] was revealed between the two conditions (Fig. 3j). Remarkably, all of these phenotypic traits perdured until the late timepoint (Fig. 3j).

Altogether, based on scRNAseq analyses, p16^Ink4a Hi senescent cells are a small subset of malignant cells. Their partial removal impacts the remaining malignant cells leading to a long-lasting switch to a more oligodendroglial-like phenotype and a decrease in the expression of genes signatory of mesenchymal cell identity.

## Modulation of the immune compartment following the partial removal of p16^Ink4a Hi cells

The mesenchymal transcriptional GBM subtype is associated with enhanced expression of anti-inflammatory and tumor-promoting macrophages[29,31,44]. We, therefore, examined the immune compartment in the scRNAseq data at the early timepoint following the partial removal of p16^Ink4a Hi cells. UMAP clustering of *CD45*+ cells revealed seven clusters in the two experimental conditions (Fig. 3d, Supplementary Fig. 3d and Fig. 4a). Differentially expressed (DE) genes and GSEA allowed the labeling of these clusters into infiltrating bone marrow-derived macrophages (BMDM), resident microglia and T cells[45] (Fig. 4b and c, Supplementary Fig. 4a; Supplementary Data 1). All the BMDM and microglia clusters harbored an anti-inflammatory gene signature. Furthermore, the BMDM-like1 and microglia clusters

shared an antagonist pro-inflammatory gene signature[32] (Fig. 4c; Supplementary Data 1). In addition, the proportion of the immune fraction within the tumor hardly varied between WT+GCV and p16-3MR+GCV GBMs. However, the number of T cells increased (from 9% to 27%), whereas the number of BMDM decreased (from 41% to 30%) upon the partial removal of p16^Ink4a Hi cells (Fig. 4d). This latter phenotype was confirmed by GSEA on bulk RNAseq data, which showed a significant decrease in transcripts associated with a core BMDM signature at the early and late timepoints in p16-3MR+GCV GBMs compared with controls (Supplementary Fig. 4b). In addition, an estimation of the abundances of the main immune cell types from our bulk RNAseq data using CIBERSORT pointed to a significant decrease of BMDM upon partial removal of p16^Ink4a Hi cells at the late timepoint (Supplementary Fig. 4c). Further, qPCR analysis on bulk RNA at the late timepoint showed significantly decreased expression levels of *Gda* and *Crip1*, two BMDM markers[32] in treated GBMs compared with controls whereas expression levels of *Tmem119* and *P2ry12*[32] two microglia markers, did not vary between both conditions (Supplementary Fig. 4d).

We next examined whether the activity of immune cell types was altered in GBM tumors partially depleted of senescent cells. GSEA on scRNAseq data at the early timepoint revealed an upregulation of TNFA signaling via the NFKB pathway in the microglia cluster and a downregulation of genes associated with the epithelial-to-mesenchymal transition, inflammatory and hypoxia pathways in the BMDM clusters in p16-3MR GBMs compared with WT+GCV GBMs (Fig. 4e). Close examination of the DE genes in these pathways revealed a significant increase in the expression of genes associated with a pro-inflammatory signature (*Ccl4*, *Tnf*, *Il1a*, *Il1b*, *Csf1*) in the microglia cluster and a significant decrease in the expression of genes related to an anti-inflammatory signature (*Cxcl2*, *Vegfa*, *Tgfbi*, Spp1, *Thbs1*, *Hmox1*, *Hif1a*) in the BMDM clusters (Fig. 4f, Supplementary Data 2). In addition, T cell cluster analysis revealed a decrease in the expression of genes regulating the activity of T cells, including the immune checkpoint genes, *Ctla4*, *Lag3*, and *Pdcd1* (encoding PD1) (Fig. 4f). Consistent with these data, GSEA of bulk RNAseq data revealed a decrease in transcripts associated with an anti-inflammatory pathway following the partial removal of p16^Ink4a Hi cells at the early and late timepoints (Supplementary Fig. 4b).

Collectively, the transcriptomic analysis at single-cell and bulk levels shows that the partial removal of p16^Ink4a Hi malignant cells modulates the abundance and the activity of tumor-associated macrophages.

## Identification of NRF2 activity and its putative targets in p16^Ink4a Hi malignant cells

To explore the regulators of senescence in p16^Ink4aHi malignant cells, we performed pathway enrichment analysis with the ENCODE and ChEA

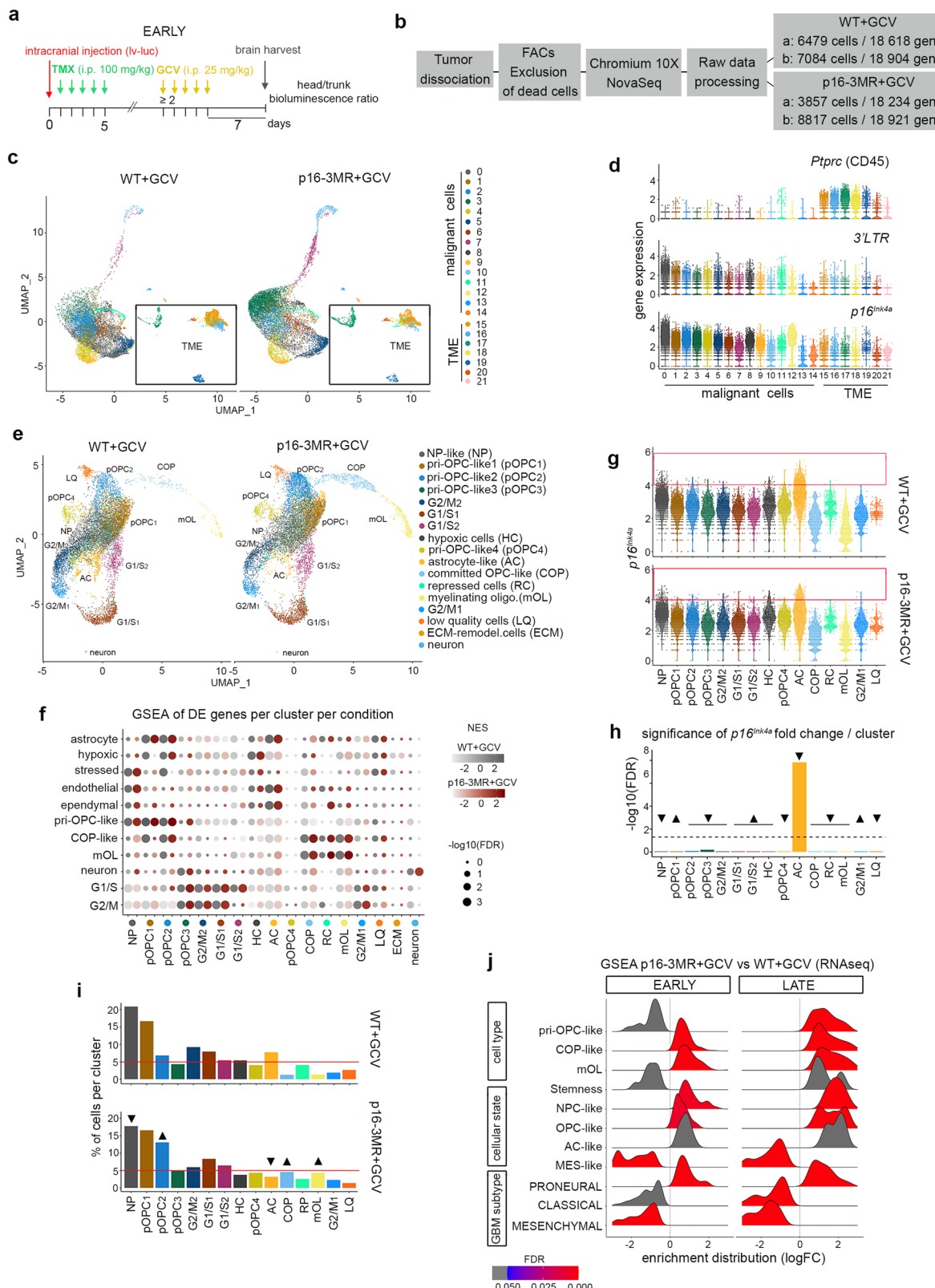

consensus TFs from ChIP-X database using Enrichr[46] on three gene sets enriched in p16[Ink4a Hi] senescent cells: (i) differentially downregulated genes in the p16-3MR+GCV vs. the WT+GCV astrocyte cluster from the scRNAseq data at the early timepoint (Early; Fig. 5a and b; Supplementary Data 3); (ii) differentially upregulated genes in p16[Ink4a] positive vs. p16[Ink4a] negative malignant cells from scRNAseq analysis at the late timepoint (Late (1); Fig. 5c and d, Supplementary Fig. 5a–e;

Supplementary Data 3); (iii) differentially downregulated genes in the p16-3MR+GCV vs. the WT+GCV GBMs from the bulk RNAseq data at the late timepoint (Late (2); Fig. 5e and f; Supplementary Data 3). Remarkably, the *Nuclear Factor Erythroid 2 Like 2* (*Nfe2l2*) signaling pathway was enriched in the three gene sets. NRF2 encoded by *Nfe2l2* is an antioxidant defense system that appears to be a plausible candidate to trigger the pro-tumoral action of p16[Ink4aHi] senescent cells as it

**Fig. 3 | Identification of p16^Ink4a Hi cells in a subset of malignant cells. a** Timeline of the mouse GBM generation for scRNAseq at the early timepoint. **b** Scheme of the scRNAseq experiment. **c** UMAP plots of WT+GCV (*n* = 2) and p16-3MR+GCV (*n* = 2) GBM cells at a 0.5 resolution and annotated malignant cells and TME cells. **d** Violin plots of the expression of *CD45*, *3′LTR*, and *p16^Ink4a* in WT+GCV GBM cells per cluster. **e** UMAP plots of WT+GCV (*n* = 2) and p16-3MR+GCV (*n* = 2) GBM malignant cells and annotated cell type at a 0.6 resolution. **f** GSEA dot plots of DE genes (FDR < 0.05; avlogFC>0.25) in WT+GCV (gray dots) and p16-3MR+GCV (red dots) GBMs of gene lists from Weng et al.[42] (Supplementary Data 1). **g** Violin plots of the expression of *p16^Ink4a* in malignant cells per cluster. The red box indicates the cells with an expression of *p16^Ink4a* ≥ 4 (p16^Ink4a Hi cells). **h** Bar plots representing the significance of *p16^Ink4a* fold change per cluster in p16-3MR+GCV GBMs compared

with WT+GCV GBMs. The arrowheads point to a decrease (arrowheads down) or increase (arrowheads up) in the fold change. **i** Bar plots representing the percentage of malignant cells per cluster in WT+GCV and p16-3MR+GCV GBMs. The arrowheads point to clusters whose cell number varies between the two conditions. **j** GSEA ridge plot of gene lists from Weng et al.[103], Neftel et al.[23], and Wang et al.[29] (Supplementary Data 1) between p16-3MR+GCV and WT+GCV GBMs at the early and late time points. Analysis performed from bulk RNAseq data. TMX tamoxifen; GCV ganciclovir; vhc vehicle; i.p. intraperitoneal; lv-luc lentivirus-luciferase; TME tumor microenvironment, UMAP uniform manifold approximation and projection, LTR long terminal repeat, DE differentially expressed, GSEA gene set enrichment analysis, FDR false discovery rate, NES normalized enrichment score, r. enrichment score running enrichment score. Raw data are provided as a Source Data file.

induces cellular senescence in fibroblasts[47] and confers a selective advantage in cancer cells[48]. Among the identified NRF2 putative targets, three genes were common to all three data sets (*Tgif1*, *Plaur*, *Gja1*) whereas eight genes were shared between two of the three gene lists (*Dap*, *Esd*, *Lmna*, *Areg*, *Igfbp3*, *Cdkn2b*, *Tnc*, and *Peak1*) (Fig. 5g; Supplementary Data 3). As illustrated on the heatmap, the combined expression of *Nfe2l2* and 11 putative target genes were unique to p16^Ink4a Hi cells in WT+GCV GBMs (Fig. 5h).

Immunohistochemistry on GBM cryosections collected at the late timepoint revealed that NRF2 was expressed in a few scattered cells (Fig. 5i and Supplementary Fig. 5f). Quantification of the NRF2 expression area in the tumor showed a modest decrease in p16-3MR +GCV compared with WT+GCV GBMs (Fig. 5j). Of note, the expression of NRF2 in cells expressing low levels of *p16^Ink4a*, most probably CD45+ cells, may have concealed decreased NRF2 expression in senescent cells (Fig. 5h). We further examined the expression of three NRF2 putative target genes whose encoded proteins are associated with senescence, glioma progression or glioma resistance, respectively, namely urokinase plasminogen activator receptor (uPAR) encoded by *Plaur*[49], Tenascin-C (TNC)[50] or Connexin43 (CX43) encoded by *Gja1*[51]. These proteins were expressed in a few scattered cells throughout the tumor. TNC was expressed in more cells than uPAR and CX43 in line with their transcript expression at the single cell level (Fig. 5h and i, Supplementary Fig. 5f). Quantification of CX43 by IHC and TNC by western blot revealed a significant downregulation of these proteins in p16-3MR+GCV GBMs compared with WT+GCV GBMs, strengthening *Gja1* and *Tnc* as NRF2 target genes in GBM (Fig. 5k and l, Supplementary Fig. 5g). We then assessed whether interactions between NRF2 selected targets and the immune fractions were observed in GBMs. We interrogated for ligand–receptor interactions between cluster 0, enriched in p16^Ink4a Hi cells, and the immune clusters in the scRNAseq data at the early timepoint using CellPhoneDB (Fig. 3d, Supplementary Fig. 5h). In silico analysis highlighted possible interactions between TNC and PLAUR, expressed in malignant cells, and integrins receptors expressed in the immune clusters. Remarkably, putative TNC-aVb3 and PLAUR-aVb3 ligand–receptor interactions between malignant cells and T cells were abolished upon partial removal of p16^Ink4a Hi cells (Supplementary Fig. 5h).

Altogether these data identify NRF2 activity and its putative target genes in p16^Ink4a Hi senescent cells and suggest that this signal could in part trigger the detrimental action of senescent cells during gliomagenesis.

## Knockdown of NRF2 in malignant cells recapitulates most features of the senolytic treatment

NRF2 has pleiotropic actions depending on cellular context. Tumor-suppressing effects of NRF2 are mediated via the maintenance of a functional immune system[48]. For instance, in a mouse lung cancer model, NRF2 activity in immune cells contributes to suppressing tumor progression[52]. To directly test whether NRF2 triggers the tumor-promoting action of malignant senescent cells, we used a knockdown approach, introducing a

microRNA targeting NRF2 (miR-NRF2) into the lentivirus used to induce gliomagenesis. We analyzed the resultant tumors at the late timepoint (Fig. 6a and b, Supplementary, Fig. 6a). Quantification of NRF2 by IHC revealed a significant decrease of the protein in miR-NRF2-GBMs compared with miR-control (ctl)-GBMs (Fig. 6c and d, Supplementary Fig. 6d). Notably, NRF2 is also expressed in CD45+ cells, not targeted by our approach, which persisted in miR-NRF2-GBMs. We performed bulk RNAseq and GSEA of miR-NRF2- and miR-ctl- GBMs at the late timepoint, and found a significant downregulation of canonical NRF2 targets and NRF2 targets from the combined analysis, confirming knockdown of NRF2 using our miR-NRF2 (Fig. 6e; Supplementary Data 3).

We next asked whether the knockdown of NRF2 in malignant cells impacted cellular senescence. The percentage of the tumor area encompassing SA-β-gal cells was similar in miR-NRF2 GBMs compared with miR-ctl GBMs, suggesting that NRF2 knockdown in malignant cells does not induce the death of senescent cells (Fig. 6f and g, Supplementary Fig. 6e). However, GSEA performed on bulk RNAseq data from miRNRF2- and miR-ctl-GBMs revealed a significant downregulation of SASP genes associated with senescence (Fig. 6e). Among these genes, *Mmp1a*, *Mmp3*, *Mmp10*, *Plau*, *Col1a2*, *Timp1*, and *Thbs1* were differentially expressed between the two conditions (Supplementary Fig. 6b). This result strongly suggests that NRF2 regulates directly or indirectly the expression of SASP genes. Further, we examined whether NRF2 knockdown mimicked the phenotype of senolytic treatment. GSEA revealed a significant decrease in the expression of genes associated with a mesenchymal identity, a BMDM signature, and anti-inflammatory pathways, similar to the gene signature changes observed in p16-3MR+GCV GBMs (Fig. 6e; Supplementary Fig. 6c). However, genes associated with an oligodendroglial identity were not modulated upon NRF2 knockdown (Fig. 6e), in contrast to the effect of the partial removal of p16^Ink4a Hi senescent cells.

Finally, as NRF2 activity protects against DNA-damaging agents and prevents carcinogenesis[53], we explored whether NRF2 knockdown impacted the onset of tumorigenesis. Live bioluminescence imaging showed no difference in the onset of tumorigenesis between the two tumor types (Fig. 6h). Most importantly, the presence of miR-NRF2 in malignant cells significantly increased the survival of GBM-bearing mice compared with controls (Fig. 6i), an effect that was more marked than upon the partial removal of p16^Ink4a Hi senescent cells (Figs. 2c, d, and 6i). One major difference between the two paradigms was the decrease of transcripts linked to the cell cycle, which occurred only in the knockdown of NRF2 in malignant cells (Fig. 6e and Supplementary 6c). One reason for this difference could be that all malignant cells were targeted by the miR strategy, whereas the p16-3MR+GCV paradigm only partially removed senescent cells.

Collectively our results show that NRF2 knockdown recapitulates most features of senolytic treatment and strongly supports NRF2 as a cellular senescence regulator in malignant cells.

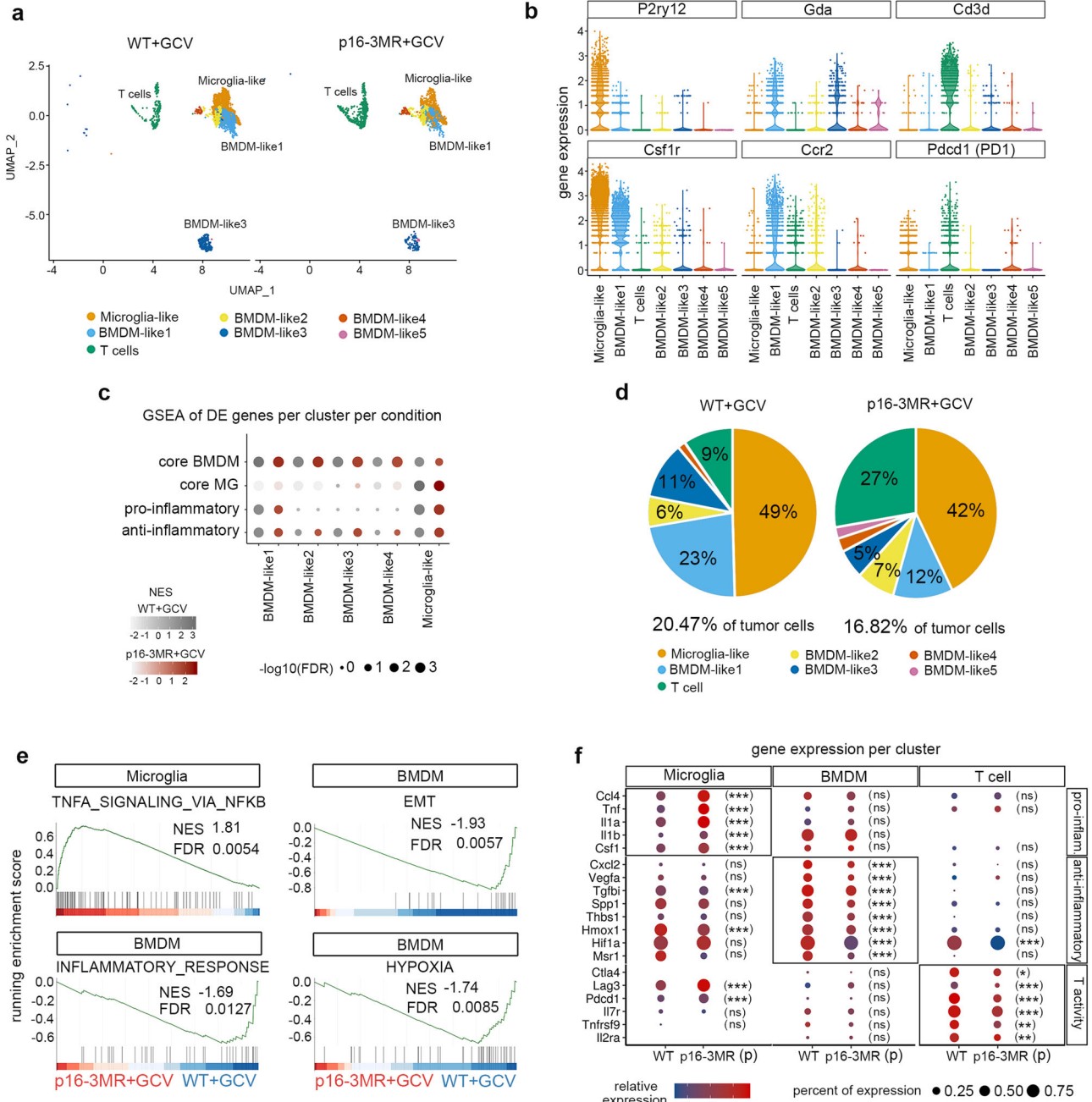

**Fig. 4 | Modulation of the immune compartment following p16^Ink4a Hi cells partial removal. a** UMAP plots of *CD45+* cells in WT+GCV and p16-3MR+GCV GBMs at a 0.5 resolution and annotated cell type. **b** Violin plots representing the expression of selected DE genes (FDR < 0.05; avlogFC > 0.25) per cluster in WT +GCV GBMs. **c** GSEA dot plot of DE genes (FDR < 0.05; avlogFC>0.25) in WT+GCV (gray dots) and p16-3MR+GCV (red dots) *CD45+* clusters of core-BMDM, core-MG, pro-inflammatory and anti-inflammatory pathways as defined in Bowman et al.[45] and Darmanis et al.[32] (Supplementary Data 1). **d** Chart pies representing the percentage of *CD45+* cells per cluster in WT+GCV and p16-3MR+GCV GBMs. **e** GSEA graphs representing the enrichment score of Hallmark gene lists in p16-3MR+GCV compared with WT+GCV microglia clusters and pooled BMDM clusters. The

barcode plot indicates the position of the genes in each gene set; red represents positive Pearson's correlation with p16-3MR+GCV expression and blue with WT +GCV expression. **f** Dot plots of the relative expression of selected genes in WT +GCV and p16-3MR+GCV microglia, pooled BMDM and T cells clusters. Statistical significance of the expression of genes in p16-3MR+GCV compared with WT+GCV clusters was determined by the Wilcoxon–Mann–Whitney test (ns, not significant, *p < 0.05; **p < 0.01; ***p < 0.001). UMAP uniform manifold approximation and projection, BMDM bone marrow-derived macrophages, MG microglia, DE differentially expressed, EMT epithelial to mesenchymal transition, GSEA gene set enrichment analysis, FDR false discovery rate, NES normalized enrichment score. Raw data are provided as a Source Data file.

## Mouse senescent signature is conserved in patient GBMs and its enrichment score is predictive of a worse survival

We next explored whether p16^Ink4a Hi senescent cells are conserved in patient GBMs. We first established a senescent signature from scRNAseq data at the early timepoint (Fig. 3). We compared the transcriptome of p16^Ink4a Hi cells in astrocyte and NP-like clusters

(p16^Ink4a Hi group) with the remaining malignant cells in WT+GCV GBMs (Figs. 3g and 7a and b). GSEA in the p16^Ink4a Hi group revealed downregulation of cell cycle pathways and an oligodendroglial state and increased expression of genes associated with inflammation, NRF2 signaling, MES-like state, mesenchymal transcriptional GBM subtype (Supplementary Fig. 7a and b). We further selected a list of 31

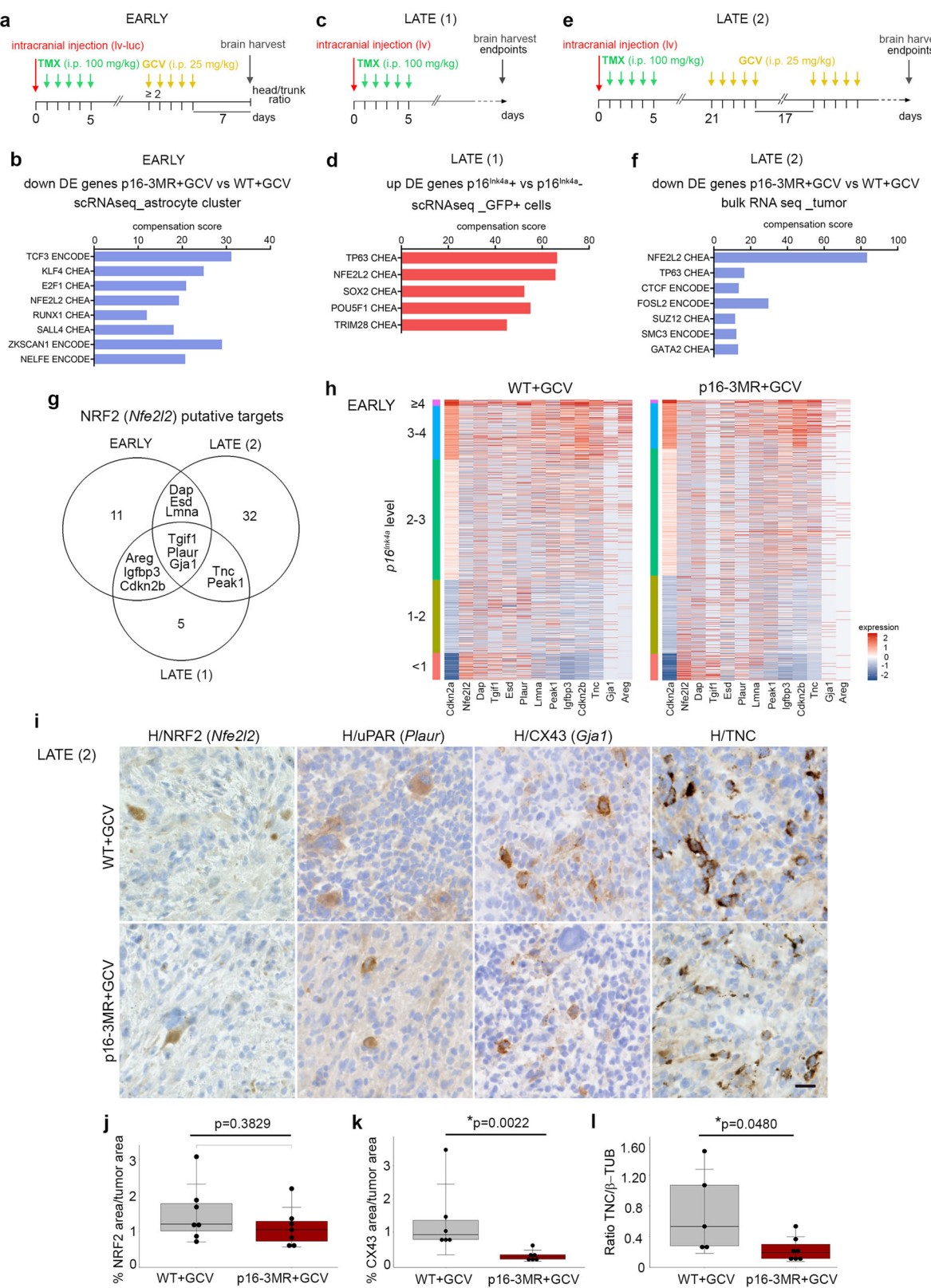

genes to define a GBM senescence signature. Among the 278 differentially upregulated genes (FDR < 0.05) in the p16[Ink4a Hi] group, we selected genes that were expressed in more than 90% of p16[Ink4a Hi] cells and presented a log2-fold change superior to 0.8 between the two groups. As expected, senescence-associated genes were enriched in the astrocyte cluster and to a lesser extent in the NP-like cluster (Fig. 7c). The encoded proteins were associated with diverse cellular processes compatible with cellular senescence, such as cell cycle arrest (*Cdkn1a*, *Cdkn2a*, *Cdkn2b*), lysosomal function (*Ctsb*, *Ctsd*, *Ctsl*, *Ctsz*, *Lamp1*, *Lamp2*), cellular growth (*Igfbp2*, *Igfbp3*), extracellular matrix interaction (*Sparc*, *Tnc*, *Sdc4*, *Lgals1*, *Timp1*, *Mt1*), cytoskeleton interaction (*Pdlim4*, *S100a11*, *Tmsb4x*, *Sep11*) and cancer (*Tm4sf1*, *Ociad2*, *Emp3*) (Fig. 7c). We then computed a single-sample GSEA (ssGSEA) senescent Z-score corresponding to the enrichment Z-score of the 31

**Fig. 5 | Identification of NRF2 activity and its putative targets in p16^Ink4a Hi malignant cells. a** Timeline of the mouse GBM generation for scRNAseq at the early timepoint (EARLY). **b** Barplot corresponding to significantly enriched pathways (ENCODE and ChEA consensus TFs from ChIP-X, Enrichr) in differentially downregulated genes (FDR < 0.05; avlogFC>0.25) in the p16-3MR+GCV compared with the WT+GCV astrocyte clusters from the scRNAseq data (as shown in **a**). **c** Timeline of the mouse GBM generation for scRNAseq at the late timepoint (LATE (1)). **d** Barplot corresponding to significantly enriched pathways in differentially up-regulated genes (FDR < 0.05; logFC > 0.5) in p16^Ink4a positive vs. p16^Ink4a negative malignant cells from the scRNAseq data (as shown in **c**). **e** Timeline of the mouse GBM generation for bulk RNAseq at the late timepoint (LATE (2)). **f** Barplot corresponding to significantly enriched pathways in differentially down-regulated genes (FDR < 0.05; logFC > 0.5) in p16-3MR+GCV compared with WT+GCV GBMs from the bulk RNAseq data (as shown in **e**). **g** Venn diagram of NRF2 putative targets between the 3 gene sets as shown in (**a**, **c**, and **e**). **h** Heatmaps of *Nrf2* and its 11 identified putative targets in WT+GCV and p16-3MR GBMs. Cells are classified into five

categories according to *p16^Ink4a* expression levels. **i** Representative immunohistochemistry (IHC, brown) counterstained with hematoxylin (H, purple) on mouse GBM cryosections at the late timepoint. H hematoxylin. (NRF2: WT+GCV, *n* = 7; p16-3MR+GCV *n* = 7; uPAR: WT+GCV, *n* = 3; p16-3MR+GCV, *n* = 3; CX43: WT+GCV, *n* = 5; p16-3MR+GCV, *n* = 6; TNC: WT+GCV, *n* = 5; p16-3MR+GCV, *n* = 7 independent mouse GBMs). Scale bar: 20 μm. **j** Quantification of the NRF2 area (IHC) over the tumor area (WT+GCV, *n* = 7; p16-3MR+GCV, *n* = 7 independent mouse GBMs). **k** Quantification of the CX43 area (IHC) over the tumor area (WT+GCV, *n* = 6; p16-3MR+GCV, *n* = 6 independent mouse GBMs). **l** Quantification of the ratio of TNC over β-TUBULIN expression (western blot) (WT+GCV, *n* = 5; p16-3MR+GCV, *n* = 7 independent mouse GBMs). Raw data are shown in Supplementary Fig. 5g. **j**–**l** data are presented as the mean ± SD. Statistical significance was determined by the Wilcoxon–Mann–Whitney test (*$p < 0.05$). i.p. intraperitoneal, lv lentivirus, lv-luc lentivirus-luciferase, TMX tamoxifen, DE differentially expressed GCV ganciclovir, H hematoxylin. Raw data are provided as a Source Data file.

genes of the senescence signature in all malignant cells of WT+GCV and p16-3MR+GCV GBM transcriptomes. As expected, the astrocyte cluster in WT+GCV GBMs contained the largest high senescent *Z*-score distribution rate (Supplementary Fig. 7c). For unbiased analysis, we defined the high distribution rate as the highest decile. This percentage dropped in all clusters in p16-3MR+GCV GBMs compared with WT +GCV GBMs, as predicted by the partial removal of p16^Ink4a Hi senescent cells (Supplementary Fig. 7c).

We then applied the ssGSEA senescence *Z*-score to three single-cell data sets of patient GBMs[23,26,54]. We analyzed separately the Neftel dataset according to the sequencing technology (Smartseq2 (SS2) and 10X). The range of the ssGSEA senescence *Z*-score of cells from patient GBMs was similar to those identified in mouse GBMs (Fig. 7d; Supplementary Fig. 7c). Hence, similar senescent transcriptomic profiles were observed in cells from mouse and patient GBMs. GBMs from the three data sets contained high senescent *Z*-score distribution rates, with the exception of 2/31 tumors, possibly due to the small number of cells sequenced in these samples (MGH126: 201 cells and MGH151: 151 cells). In summary, this analysis strongly suggests that the mouse senescent signature is conserved in cells from patient GBMs.

To assess whether the ssGSEA senescence score could be used as a prognostic factor for patients with GBM, we interrogated The Cancer Genome Atlas (TCGA) GBM data sets and performed a Cox regression analysis with five variables. Cellular senescence was linked to aging and our mouse senescence signature was based on the expression of p16^Inka. Therefore, we used as variables the ssGSEA senescence score, p16^Ink4a copy number alteration status, and the age of the patient, in addition to the Karnofsky score and sex variables commonly used for patients with GBM (Supplementary Data 4). Multivariate Cox regression model showed that regardless of p16^Ink4a status, the age of the patient, the sex, and the Karnofsky score, the enrichment of the senescence score predicted a worse survival (hazard ratio above 1) in patients with GBM (Fig. 7e; Supplementary Data 4).

Finally, we tested whether NRF2 activity could account for some of the tumor-promoting action of cellular senescence in patient GBMs, similar to mouse GBMs. First, SA-β-gal staining coupled with IHC on cryosections revealed the expression of NRF2, TNC, and CX43 in SA-β-gal+ cells in patient GBM samples (Fig. 7f and Supplementary 7d). As described above, we next interrogated ssGSEA NRF2 target scores in TCGA GBM data sets and performed a Cox regression analysis. NRF2 targets corresponded to the 59 genes identified in the combined analysis (Supplementary Data 3; Fig. 5g). The Cox regression model showed that regardless of p16^Ink4a status, the age of the patient, the sex, and the Karnosky score, an enriched NRF2 target gene score predicted worse survival in patients with GBM (Fig. 7g; Supplementary Data 4).

In summary, our data show that cells enriched for the mouse senescent signature are present in patient GBMs and that the

enrichment scores of senescence and of NRF2 targets are correlated with worse survival in patients with GBM.

## Discussion

Depending on the context, cellular senescence plays both beneficial and detrimental roles during tumor progression. Here, we revealed the tumor-promoting action of malignant senescent cells in mouse and patient GBMs. The mouse MES-GBM model used in the present study, even though its genetics differ from the patient GBMs, recapitulated the histopathology, the heterogeneity of cellular states, the infiltration of BMDM specific to the mesenchymal transcriptional GBM subtype, and the senescent features of patient GBMs (see also ref. [44]). Partial removal of p16^Inka Hi malignant senescent cells modified the tumor ecosystem and improved the survival of GBM-bearing mice. The difference in survival following a senolytic treatment appeared to be relatively modest, nonetheless, this difference was significant and was observed in the p16-3MR paradigm when compared to a first control cohort (WT+GCV). These results were repeated using a second control cohort (p16-3MR+vhc). In addition, we observed a benefic effect of another senolytic, ABT263, in a treated cohort compared to a control one. This finding is remarkable given the fact that senescent cells represented <7% of the tumor and that their removal using the p16-3MR transgene was only partial. These findings suggest that senolytic drug therapy may be a benefit for patients with GBM.

By combining single-cell and bulk RNA sequencing, immunohistochemistry, and genetic knockdowns, our study established a link between senescence and NRF2 activity in the context of GBM. Chronic activation of NRF2 contributes to tumor growth, metastasis, treatment resistance, and poorer prognosis in patients with cancer[48]. Depending on the context, NRF2 promotes or delays fibroblast senescence[47,48]. NRF2 binds to antioxidant-responsive elements (AREs) and controls the expression of a battery of genes regulating metabolism, intracellular redox balance, apoptosis, and autophagy[48]. Under a homeostatic state, cytoplasmic NRF2 binds to KEAP1, which mediates its proteasomal degradation. Impairment of NRF2-KEAP1 binding, either by phosphorylated-p62/SQSTM1 or by elevated ROS permits NRF2 nuclear translocation and subsequent activation of target genes[55–57]. Of note, p62/SQSTM1-mediated degradation of KEAP1 and NRF2 promotes in vitro glioma stem cell survival[58]. NRF2 is hyperactivated preferentially in the MES-GBM subtype[58] however, *NRF2/NFE2L2* and genes regulating its activity (*KEAP1, SQSTM1, CUL3, RBX1, SKP1, CUL1, BTRC, SYVN1*) are rarely altered in GBMs (see cbioportal.org). The putative triggers of senescence in our GBM model and in patient GBMs are also known to regulate NRF2 activity such as hypoxia, ROS, PI3K–AKT pathway (enhanced by the loss of PTEN)[48,59] or *Nrf2* transcription such as RAS oncogene (K-RAS)[60]. Future work is required to establish the contribution of these triggers in the process of cellular senescence and NRF2 activity in GBMs. In the present study, we identified NRF2

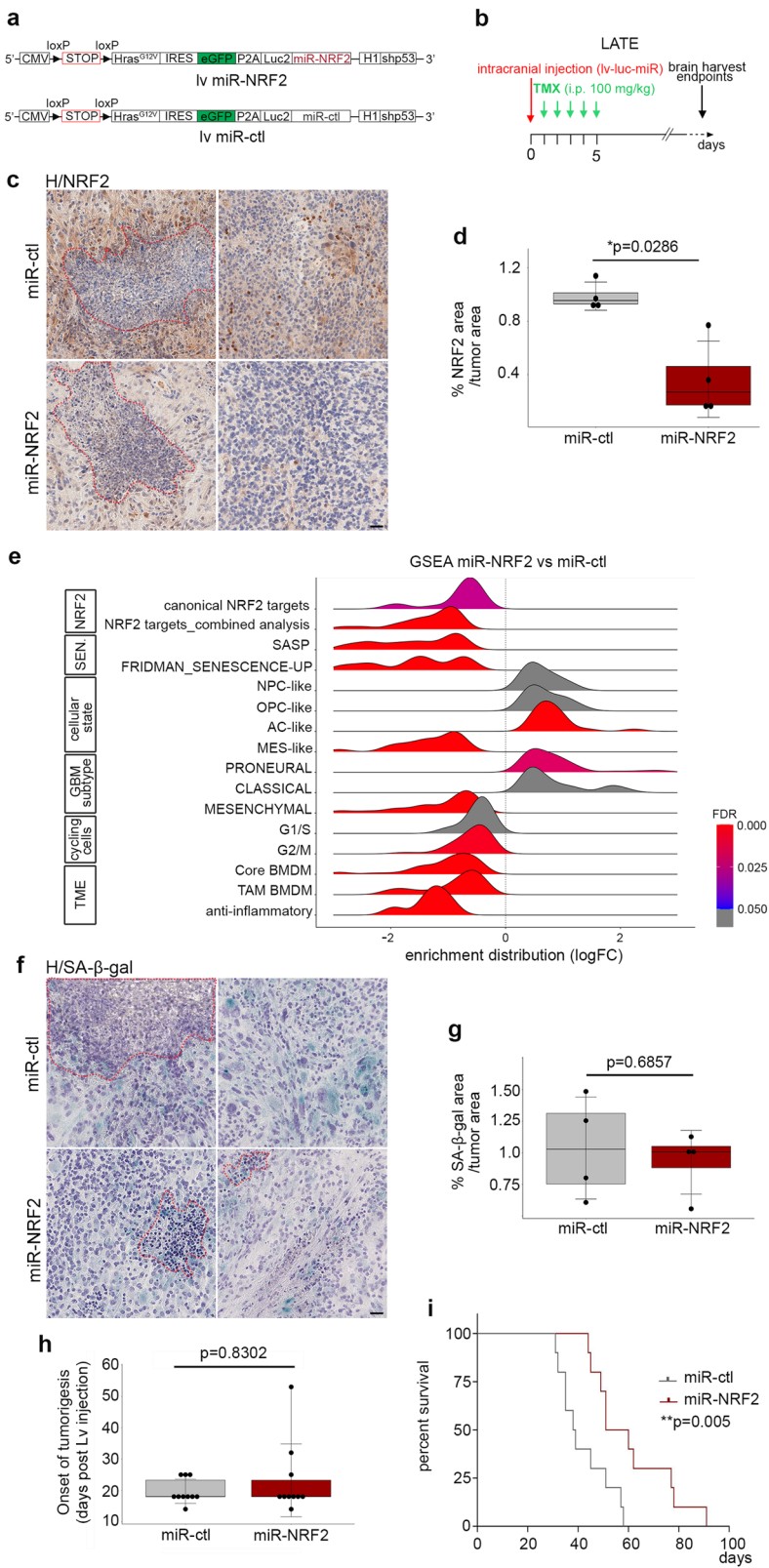

putative targets that are not canonical NRF2 targets. These targets encode for growth factors (AREG, IGFBP3, TGIF1), ECM components or remodelers (TNC, uPAR, ESD), or cell–cell interactors (CX43) and have been previously identified as SASP factors[39–41]. TNC and CX43 are of particular interest. Indeed, CX43 participates in the formation of microtubes that interconnect malignant cells, creating a cellular network resistant to treatment[51]. Further, the pro-tumoral functions of

TNC have been described, independently of the senescence context[61,62]. TNC is a component of the glioma ECM that binds to integrin receptors, EGF receptor (EGFR), and SYNDECAN 4 (SDC4) and regulates angiogenesis, proliferation, and cell migration[50]. Hence, TNC functions could partly be responsible for the tumor-promoting phenotype of the p16[Ink4a Hi] malignant senescent cells. As mentioned above, NRF2 has pleiotropic actions depending on the cellular context and is

**Fig. 6 | Knockdown of NRF2 in malignant cells recapitulates most features of the senolytic treatment. a** Scheme of the lentiviral vector containing either a miR-NRF2 or a miR-ctl. **b** Timeline of the mouse GBM generation at the late timepoint. **c** Representative NRF2 IHC staining (brown) on miR-ctl ($n = 4$) and miR-NRF2 ($n = 4$) GBM cryosections. Necrotic areas are outlined in red dashed lines. **d** Quantification of the NRF2 area (IHC) over the tumor area (miR-ctl $n = 4$; miR-NRF2 $n = 4$). **e** GSEA ridge plot on bulk RNAseq of miR-NRF2-GBMs compared with miR-ctl-GBMs (see Supplementary Data 1 for gene lists). **f** Representative SA-β-gal (blue) staining on miR-ctl ($n = 4$) and miR-NRF2 ($n = 4$) GBM cryosections. Necrotic areas are outlined in red dashed lines. **g** Quantification of the SA-β-gal area over the tumor area (miR-

ctl $n = 4$; miR-NRF2 $n = 4$). **h** Boxplot representing the onset of tumorigenesis in miR-ctl ($n = 10$) and miR-NRF2 ($n = 10$) mice following post-lentiviral injection. The onset of tumorigenesis is defined when the bioluminescence reached 3e10⁶. **i** Kaplan–Meier survival curves of miR-ctl ($n = 10$, median survival 38.5 days) and miR-NRF2 mice ($n = 10$, median survival 55.5 days). Statistical significance was determined by the Mantel−Cox log-rank test (**$p < 0.01$). Scale bar, **c** and **f** 50 µm. **d**, **g**, **h** Statistical significance was determined by Wilcoxon–Mann–Whitney test (*$p < 0.05$). lv lentivirus, miR-ctl miR-control, H hematoxylin, GSEA gene set enrichment analysis, sen. senescence, TAM-associated macrophages, BMDM bone marrow-derived macrophages. Raw data are provided as a Source Data file.

expressed in multiple cell types. Of note, in this study, we did not address the function of NRF2 in the GBM microenvironment, notably in CD45+ cells. To sum up, our findings suggest that a senolytic treatment may represent a therapeutical strategy to eliminate NRF2+ malignant senescent cells without targeting NRF2+ cells in the microenvironment.

Single-cell RNAseq analysis of mouse GBMs allowed the comprehensive characterization of the pro-tumorigenic malignant senescent cells. Although our approach focused primarily on p16[Ink4a] Hi senescent cells in a mouse MES-GBM model, our findings show that the senescence signature we established in this study, is applicable to GBMs regardless of p16[Ink4a] status (Fig. 7e). The presence in the senescence signature of three genes (*Cdkn2a*, *Cdkn2b*, *Cdkn1a*) encoding for cyclin-dependent kinase inhibitors warrants cell cycle exit and entry into senescence. Of note, *CDKN1A* (*p21[CIP1]*) is rarely mutated in patient GBMs (0.4%) and p21[CIP1] mediates senescence in many tissues[3]. Furthermore, we presume that the senescent signature defined in the study is specific to detrimental senescence. Indeed, the enrichment of the senescence score predicts a worse survival in patients with GBM and multiple genes in the signature encode for proteins whose activities are associated with tumor aggressiveness and/or worse patient prognosis (CD151[63]; EMP3[64]; IGFBP2[65]; LGALS1[66]; TMSB4X[67]; TNC/SDC4[62]; SPARC[68]; TIMP1[69]). Future studies will determine whether the senescence scoring could be used in the diagnosis of patients with GBM to improve the design of personalized treatment and effective combinatorial strategies.

In this study, we showed that senolytic treatments applied to GBM-bearing mice delay temporarily tumor growth (Fig. 2c and d). Previous work investigated the action of the specific inhibitors of the anti-apoptotic BCL2 and BCL-xL proteins, ABT737 and ABT263, in the context of GBM mouse models. Similarly to our results (Fig. 2d), ABT737 treatment increases the survival of immunodeficient mice grafted with the human glioma cell line U-251MG[70]. ABT263 treatment when combined with drugs decreasing Mcl1 levels (a BCL2 family protein), decreased tumor volume in a heterotopic (subcutaneous) model of proneural GBM[71,72]. Furthermore, ABT263 treatment provides a survival benefit only when applied in combination with the onco-metabolite 2-R-2-hydroxyglutarate (2-HG; produced by IDH1-mutated tumors) in a proneural GBM model[73]. Cellular senescence was not addressed in these studies but in the light of our results using the p16-3MR transgene, some of the cells targeted by this approach may be malignant senescent cells, extending the notion of detrimental senescent cells to distinct GBM subtypes in agreement with the senescence score analysis performed on data from patient GBMs (Fig. 7). All together our findings and these previous studies, raise the possibility to use senotherapy to improve the outcome of a patient with GBM.

Many important issues remain to be solved before envisioning senotherapy in the context of GBM such as the nature of the senolytic, the timing of its administration (neoadjuvant-concomitant adjuvant or adjuvant), and the most effective combination of treatment. First of all, these drugs should cross the brain–blood-barrier. Novel molecules are expected to be discovered in the near future as the field of seno-therapies, which includes drugs eliminating senescent cells (senolytic drugs: anti-BCL2 and BCL-xL[13], dasatinib and quercitin[14], cardiac glycosides[74–76], CAR T cells[77]) and drugs inhibiting their function

(senostatic drugs such as Metformin[78]), is under active investigation[79]. Despite encouraging results in mouse GBM models, clinical studies show major side effects of ABT263 such as thrombocytopenia and neutropenia caused by BCL-xL inhibition[80], limiting the use of BCL-xL inhibitors as safe and effective anticancer agents. The selective BCL2 inhibitor ABT199 (Venetoclax) which spares the platelets, displays variable results on senescent glioma cells, in vitro[81,82]. Administration of a combination of dasatinib and quercitin, a multi-tyrosine kinase inhibitor and a natural flavonoid respectively, eliminates efficiently senescent cells in different pathologies in the mouse including neurodegenerative diseases[83,84]. Results of ongoing clinical trials using these drugs on cohorts of patients with mild cognitive impairment and Alzheimer's disease (NCT04785300i, NCT04685590ii) should encourage or refute this approach for targeting senescence in the central nervous system.

Heterotopic and orthotopic mouse models of cancers showed the efficacy of the one−two punch sequential therapy defined by a pro-senescence therapy, followed by senolytic therapy clearing the induced-senescent cells and preventing the accumulation of detrimental persistent senescent cells[18,77,85–87]. The efficacy of such a strategy implies a homogeneous response to the pro-senescence therapy. GBMs which are characterized by intra-tumoral heterogeneity, are thus not the ideal candidate for the use of this strategy. Nonetheless, radiation and TMZ chemotherapy induce senescence in glioma cells in vitro and sensitize these cells to anti-apoptotic inhibitors with distinct efficacy according to the patient-derived cell lines, suggesting that this strategy may be relevant for some patients with GBM[82,88]. Based on our work which focused on naturally-occurring senescence, we hypothesize that senolytics combined with conventional treatment could have a double action by depleting both resident and therapy-induced senescent cells which may potentialize their effect. Also, we would like to propose another combined therapeutical strategy that would take advantage of the tumor ecosystem modifications following senolytic treatment. For example, promyelinating drugs may amplify the plasticity of malignant cells towards an oligodendroglial-like differentiated phenotype initiated by the senolytic treatment (Fig. 3). Another strategy could be to target the immune cells. Indeed, although a thorough study on the consequence of senolytic treatment on the immune system is required, our study provides evidence of a decreased anti-inflammatory phenotype after treatment (Fig. 4). Therefore, senolytic treatment may prime GBM to respond to immunotherapy. This hypothesis is attractive as the immunotherapies with the anti-PD1 and PD-L1 antibodies did not show an extension of the overall survival in treating patients with recurrent GBM[89,90]. One possible explanation for this failure could be that GBMs contain very few immune effector cells[91]. Further work on immunocompetent GBM models, reproducing the intra-heterogeneity of patient GBMs, is now needed to evaluate the effect of senotherapy on glioma progression and assess their efficacy as companion therapy.

## Methods
### Ethics
Fresh patient GBM samples were selected from the Pitié-Salpêtrière tumor bank Onconeurotek. They were reviewed by our senior

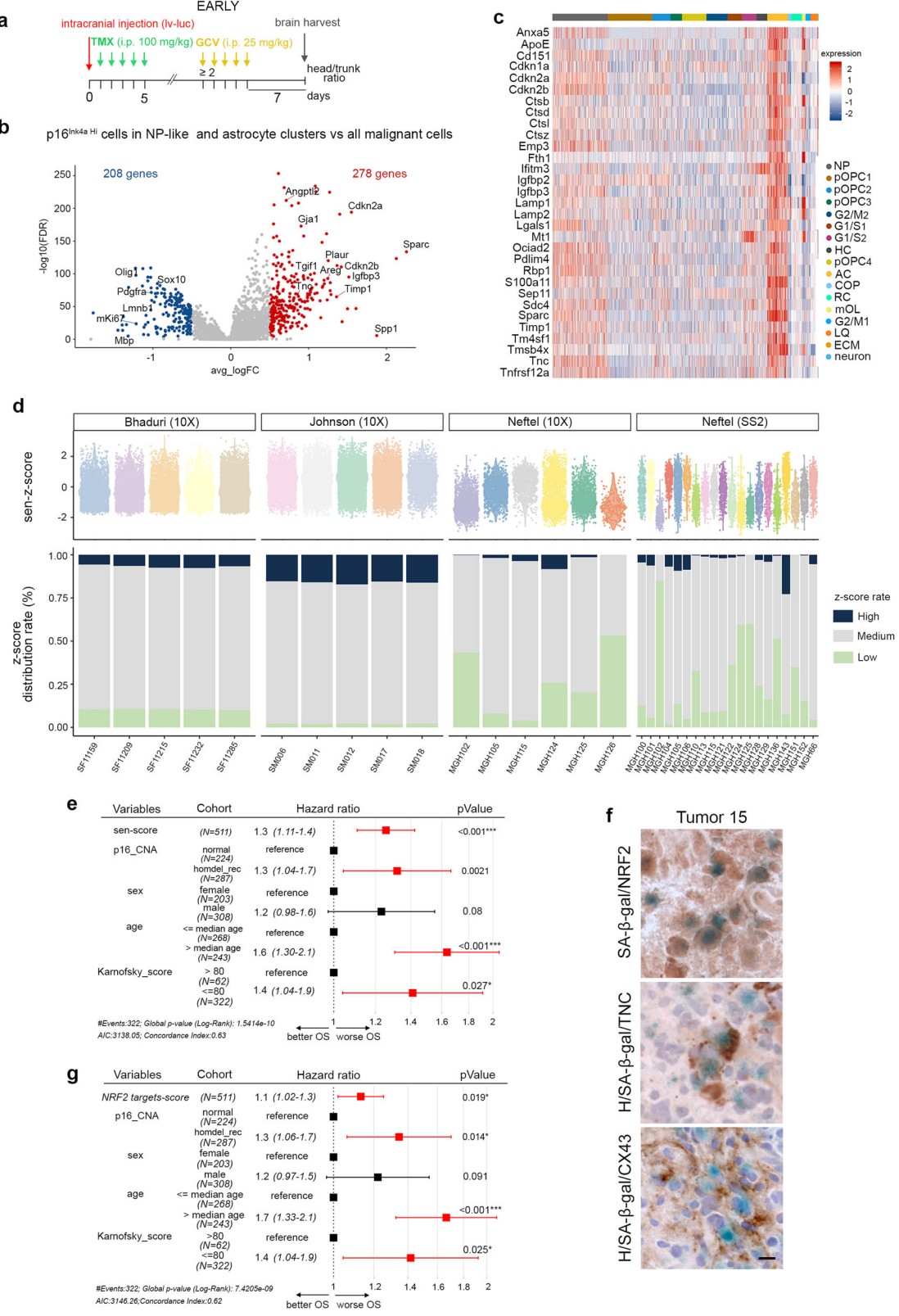

pathologist (F.B.) to validate the histological features and confirm patients' diagnoses. Collection of tumor samples and clinical-pathological information were obtained upon patients' informed consent and ethical board approval, as stated by the Declaration of Helsinki. The ethical approval was obtained from the ethical committee «CPP–Ile de France VI–Groupe Hospitalier Pitié Salpêtrière». Molecular characterizations were performed as previously described[92].

All animal care and treatment protocols complied with European legislation (no. 2010/63/UE) and national (French Ministry of Agriculture) guidelines for the use and ethical treatment of laboratory animals. All experiments on animals were approved by the ethical committee in animal experimentation Charles Darwin no. 5, Paris (approval APAFIS 9131). In the approved document the maximal tumor burden, as defined as end-points in this study, was determined by a loss of 10–15% of the mouse body weight in the 48 h interval and/or the

**Fig. 7 | Mouse senescent signature is conserved in patient GBM and its enrichment score is predictive of worse survival. a** Timeline of the mouse GBM generation for scRNAseq at the early timepoint. **b** Volcano plot of differentially expressed (DE) genes ($-0.5 <$ log2FC $> 0.5$; FDR $< 0.05$) between p16[Ink4a Hi] cells (gene expression $\geq 4$) of astrocyte and NP-like clusters compared with the remaining malignant cells in WT+GCV GBMs. **c** Heatmap of the 31 senescence signature genes in WT+GCV GBMs. **d** Top: Violin plots of the single-sample GSEA (ssGSEA) senescent *Z*-score in all patient GBM cells. Patient GBMs data were extracted from Bhaduri et al.[26], Johnson et al.[54], and Neftel et al.[23]. Bottom: Barplots of the percentage of the ssGSEA senescent *Z*-score distribution rate in all patient GBM cells. High and Low distribution rates correspond to the highest and lowest decile, respectively. **e** Table representing a Cox regression model using the ssGSEA-senescence score (sen-score), p16[INK4a] copy number alteration (p16-CNA) stratified

into a group without alteration (normal) and a group harboring homozygous recessive deletion (homdel_rec) in the *INK4a* locus, the sex, the age of the patients and the Karnofsky score. **f** Representative SA-β-gal staining (blue) coupled with IHC (brown) and counterstained with hematoxylin (H) on patient GBM cryosections. Three patient GBMs were analyzed per antibody. Scale bar: 10 µm. **g** Table representing a Cox regression model using the ssGSEA NRF2 targets score (NRF2 targets score), p16[INK4a] copy number alteration, the sex, the age of the patients, and the Karnofsky score. **e** and **g** Data were extracted from The Cancer Genome Atlas (TCGA) GBM data sets and statistical significance was determined by a Log-Rank test. The error bars correspond to 95% Confidence Intervals (CIs). GCV ganciclovir, TMX tamoxifen, i.p. intraperitoneal, lv lentivirus, lv-luc lentivirus-luciferase, OS overall survival. Raw data are provided as a Source Data file.

degradation of the general condition of the animal such as prostration/decubitus, loss of alertness, cutaneous lesions.

## Mouse breeding
To generate the GBM mouse model, we crossed Glast[creERT2/+] mice[93] with the Pten[fl/fl] mice[94]. Glast[creERT2/+]; Pten[fl/fl] males were bred with either Pten[fl/fl] or Pten[fl/fl]; p16-3MR/+females[17] to generate Glast[creERT2/+]; Pten[fl/fl] and Glast[creERT2/+]; Pten[fl/fl]; p16-3MR/+mice, named WT and p16-3MR mice respectively. All animals used in the study were from mixed genetic backgrounds C57BL/6J and Swiss and they were 6–8-week-old females except for the mice used for scRNAseq at the early timepoint that were 14-week-old females.

## Plasmid construction
H-RasV12-shp53-(IRES)-GFP-(2a)-Firefly-luciferase vector was generated from the H-RasV12-shp53-(IRES)-GFP[37] construct using the Gibson Assembly technique[95]. The terminal IRES-GFP region of the initial vector and a P2A-luciferase2 sequence were both flanked with a shared sequence overlap and amplified by PCR. They were then inserted into the SalI and PmlI sites of the initial vector.

Four oligonucleotide sequences for the miR-NRF2-based shRNAs targeting have been designed using the Block-iT RNAi designer tools (Invitrogen; Supplementary Data 5) and cloned into the pcDNA6.2-GW /EmGFPmiR plasmid according to the manufacturer protocol (BLOCK-iT polII miR RNAi, invitrogen #K4936-00). miR-NRF2 #4 and a miR-ctl[96] (Supplementary Data 5) have been further cloned by the Gibson Assembly technique in the H-RasV12-shp53-(IRES)-GFP-(2a)-Firefly-luciferase vector.

## Stereotaxic injection
We stereotaxically performed lentiviral intracranial injection of mice to induce de novo tumorigenesis. The mice were anesthetized with isoflurane (2–3%, 1 L/min oxygen), and subcutaneously injected in the head with lidocaine (60 µL, 2.133 mg/mL). Analgesia was injected intraperitoneally (i.p.) during presurgery and up to 24 h after surgery (buprenorphine, 100 µL, 15 µg/mL). The HRasV12-shp53-(IRES)-GFP (lv) or the HRasV12-shp53-(IRES)-GFP-(2a)-Firefly-luciferase lentivirus (lv-luc, 1 µL, $6 \times 10^8$ PFU/mL) was injected in the right subventricular zone (SVZ) of the brain ($x = 1$ mm, $y = 1$ mm, $z = -2.3$ mm from the bregma). We used a Hamilton 30 G needle with a silica fiber tip (MTI-FS) and an automatic injector (Harvard Apparatus). After injection, the skin wound was closed with surgical glue (SurgiBond®) and animals were placed under an infrared lamp until they recover a vigil state. From the next day, mice were injected i.p. with tamoxifen (TMX, 20 mg/mL in corn oil, Sigma #T5648-1Gi and Sigma #C8267) once per day for 5 consecutive days to induce the recombination of the *Pten* locus and of the *loxP-RFP-loxP* cassette of the lentivirus allowing the expression of *H-RasV12*.

## Bioluminescence imaging
We monitored tumor growth by in vivo bioluminescence twice a week from 14 days post intracranial injection. The mice were i.p. injected

with Xenolight D-Luciferin (100 µL, 30 mg/mL, Perkin Elmer #122799), anesthetized with isoflurane and their head and back were shaved. Bioluminescence was recorded with an IVIS Spectrum In Vivo Imaging System (Perkin Elmer) and the ratio was measured by normalizing the head signal on the back signal. The onset of tumor growth corresponds to a head/trunk bioluminescence ratio of 2 (see below) for the p16-3MR +vhc and p16-3MR+GCV mice and to a head bioluminescence signal of $3e10^6$ for miR-ctl and miR-NRF2 GBM-bearing mice. The difference in the evaluation of tumor growth was due to a point mutation in the P2A sequence in the HRasV12-shp53-(IRES)-GFP-(2a)-Firefly-luciferase vector.

## Mouse treatments
Mice were treated with vehicle (PBS, DMSO 20%, Sigma #D8418-50ML) or GCV (25 mg/kg/day, Selleckchem, #S1878) prepared in PBS, 20% DMSO at 21 days post injection (DPI). During the course of the study, we implemented bioluminescence-monitored GBM growth for the two paradigms p16-3MR+vhc vs. p16-3MR+GCV and the WT+vhc vs. WT +ABT263 (see below). The mice were treated when head to back bioluminescence ratio was superior or equal to 2 (around 24 DPI; $n = 43$). GCV was administered via daily i.p. injections for 5 consecutive days per cycle, for two cycles with a 2-week interval between the two cycles. ABT263 (Selleckchem, #S1001) was prepared as previously described[13] and was administered to mice by gavage at 50 mg/kg/day for 5 days.

## Kaplan–Meier mice survival studies
Kaplan–Meier survival analysis was done using Prism (Graphpad software v.8.2.1). In accordance with EU guidelines, mice were sacrificed when reaching endpoints (20% body weight decrease, deterioration of general condition). Mice were injected by batch. One batch always included control and experimental mice injected the same day. When control mice survival extended more than 57 DPI, the entire batch was removed from the analysis to exclude technical bias linked to intracranial injection.

## Mice brain collection
When reaching endpoints, mice were sedated with $CO_2$ inhalation followed by intracardiac perfusion with cold HBSS 1X. After harvesting the brain, the GFP+ tumor was cut into two parts, under the MZFL II stereomicroscope (Leica). The anterior part of the GFP+ tumor and the GFP− parenchyma were chopped and stored in TRI-reagent (Molecular Center Research, #TR 118) at −80 °C or directly snap-frozen in liquid nitrogen for RNA isolation. The posterior part was snap-frozen in dry ice-cooled-isopentane for histological studies. Brains were cryosectioned at a 12-µm thickness (Leica cryostat).

## SA-β-gal and immunohistochemical staining
For SA-β-gal staining, GBM sections were fixed in 2% PFA and 0.02% glutaraldehyde (Sigma, #340855) for 10 min at RT. Note, that the concentration of glutaraldehyde was dropped from 0.2% to 0.02% from the original protocol[97] to allow combined immunohistochemical

staining. Sections were washed twice in PBS pH 7.0 and once in PBS pH 5.5 for 30 min. Slides were incubated in the X-gal solution as previously described[97] for 5 h 30 min at 37 °C for mouse sections and overnight (O/N) for patient GBM sections. Slides were then washed in PBS and post-fixed in 4% PFA for 10 min at RT.

For Immunohistochemical staining, GBM sections were fixed in 4% PFA for 10 min and washed in PBS. Endogenous peroxidases were inactivated in 1% $H_2O_2$ (in $H_2O$) solution for 5 min and sections were incubated in the blocking solution (PBS 1X, 10% NGS, 3% BSA, and 0.25–0.5% Triton) for 30 min. Sections were then incubated with the primary antibody (Supplementary Data 5) in the blocking solution for either 2 h at RT or O/N at 4 °C. Slides were rinsed and incubated with biotinylated secondary antibodies for 45 min at RT. An amplification step was performed using VECTASTAIN Elite ABC HRP Kit (Vector Laboratories, #PK-6100-NB) for 30 min at RT and staining was revealed by a DAB reaction. Images were acquired using an Axio Scan.Z1 (Zeiss) and extracted using the ZEN 2.0 blue edition (Zeiss) software.

### Surface area quantification

Quantifications were performed using Fiji software (v.2.1.0/1.53c)[98]. Region of interest (ROI) corresponding to the tumor, was selected using the ellipse tool. IHC images were then color deconvoluted according to the "Giemsa" or "Hematoxylin and DAB (H DAB)" vector to assess a threshold of the SA-β-gal or DAB signal, respectively. The signal threshold was adjusted in order to remove the unspecific background signal without clearing the specific one. Number of pixels was measured and the values were normalized on the GFP+ tumor surface area. For mice brain tissue, four slides with three sections on each ($n = 12$) for SA-β-gal quantification and three slides with three sections on each ($n = 9$) for IHC quantification were analyzed per sample. For the patient sample, four sections were analyzed for SA-β-gal quantification.

### Western-blotting

Total proteins were extracted from tumor samples following the TRI-reagent protocol (Molecular Center Research, #TR 118). Protein pellets were solubilized in 1% SDS, 10 M urea, and stored at −80 °C. Protein concentration was assayed using the Pierce BCA protein assay kit (Thermo Fisher, #23225). Proteins were separated on 4–20% stain-free polyacrylamide gels (Mini-PROTEAN TGX Protein Gels, Bio-Rad, #4568096) and transferred on a nitrocellulose membrane 0.45 μm (Thermo Fisher, #88018). Membranes were probed with primary antibodies (Supplementary Data 5) diluted in Super Block Blocking buffer in TBS (Thermo Fisher, #37535) and incubated O/N at 4 °C under gentle agitation. The secondary antibodies were incubated for 1 h at RT. Fluorescence was detected using the Odyssey CLx (Li-cor), and specific bands were quantified using Fiji software (v.2.1.0/1.53c)[98] and normalized against the corresponding β-TUBULIN band.

### Cell culture

Glioma 261 murine cell line (GL261) was cultured in DMEM (Thermo Fisher #31966021) 1% fetal bovine serum (Thermo Fisher #A3160801). The GL261 cell line was not authenticated. These cells were transfected with the pcDNA6.2-GW /EmGFPmiR plasmids containing miR sequences (miR-NRF2 #1, #2, #3, #4 and miR-ctl[96] using the FUGENE HD transfection reagent (Promega #E2311). Seven days later GFP positive cells were isolated by flow cytometry (Biorad S3e cell sorter), cultured for 2 more days and their total RNAs were extracted.

### RNA extraction and RT-qPCR

Total RNAs were extracted from tumor and parenchyma samples and GL261 cells following either the TRI-reagent (Molecular Center Research, #TR 118), the Maxwell RSC simplyRNA Tissue (Promega,

#AS1340), and the Macherey-Nagel Mini kit Nucleospin protocol (Macherey-Nagel, #740955.50).

cDNAs were generated using the Maxima 1str cDNA Synth Kit (LifeTechnologies, K1642). Quantitative PCR was performed using LightCycler 480 SYBR Green I Master Mix (Roche, #4707516001) on a LightCycler® 480 Instrument II (Roche). Samples were run in duplicate or triplicate, transcript levels were normalized to TBP and GAPDH, and analysis was performed using the $2^{-\Delta\Delta CT}$ method[99]. Primers used in this study are listed in Supplementary Data 5.

### Bulk RNA-seq and analysis

The quantity and quality of the total RNAs extracted were assessed by the Tapestation 2200 (Agilent) and sequenced with the Illumina NextSeq 500 Sequencing system using NextSeq 500/550 High Output Kit v2 (150 cycles, # 20024907), 400 millions of reads, 50Gbases.

Quality of raw data was evaluated with FastQC (v.0.11.5). Poor quality sequences were trimmed or removed with Fastp software to retain only good-quality paired reads. Star (v2.5.3a) was used to align reads on mm10 reference genome using default parameters except for the maximum number of multiple alignments allowed for a read which was set to 1. Quantification of gene and isoform abundances was done with rsem (v.1.2.28) on RefSeq catalog, prior to normalization with edgeR Bioconductor package (v.3.28.0). Finally, differential analysis was conducted with the glm framework likelihood ratio test from edgeR. For malignant samples, a batch effect was detected in PCA representation. To correct it, we performed the analysis by using an additive model which includes this batch variable. Multiple hypothesis-adjusted $p$-values were calculated with the Benjamini–Hochberg procedure to control FDR.

Functional enrichment analysis was performed with clusterProfiler (v3.14.3) Bioconductor package on the differentially deregulated genes with over-representation analysis (enricher function) and on all the genes with Gene Set Enrichment Analysis (GSEA function[100]). Hallmark, Transcription factor targets (TFT), and Canonical pathways (CP) gene sets from MSigDB collections have been used, complete with some custom gene sets (Supplementary Data 1). CIBERSORT[101] was used to accurately quantify the relative abundances of six distinct immune cell types according to the ImmGen immune cell genes signature (reference GSE124829).

### Tumor dissociation for scRNAseq

After brain harvest, GFP+ tumors were dissected under a Leica MZFL II stereomicroscope. Tumor pieces were chopped and incubated for 5 min at 37 °C in an HBSS-papain-based lysis buffer (Worthington PAP) containing DNAse (0.01%, Worthington #LS002139) and L-Cystein (124 μg/mL, Sigma #C78805). Papain digestion was inhibited by ovomucoid (7 mg/mL, Worthington #LS003085). Tissue was further dissociated mechanically and centrifuged $300 \times g$, 10 min at 4 °C. Cells were resuspended in cold HBSS, a debris removal step was performed (Miltenyi #130-109-398) and blood cells were removed using a blood lysis buffer (Roche 11814 389001). After centrifugation, cells were resuspended in cold HBSS and incubated with the eBiosciences Fixable Viability Dye Fluor 450 or 660 (Invitrogen 65-0863), to label dead cells, and washed. Cells were then sorted using the MoFlo Astrios cell sorter (Beckman Coulter) or the S3e cell sorter (Biorad). Live cells were collected in HBSS 0.1% BSA precoated tubes, centrifuged, and resuspended in HBSS-0.1% BSA at a concentration of 1200 cells/μL. GFP+ and GFP− cells were collected separately for the scRNAseq ddSeq experiment.

### Single-cell RNA sequencing and analysis−ddSeq data

Cell suspension of one dissociated GBM was loaded in 4 wells (3 wells with GFP+ cells or malignant cells and 1 well with GFP− cells or non-malignant cells) on the ddSEQ Single-Cell Isolator (Biorad). A library was generated using SureCell™ Whole Transcriptome Analysis 3′

Library Prep Kit for the ddSEQ System (Illumina, #20014280) and was sequenced on a Nextseq 500 Illumina sequencing system, using a High Output Kit (150 cycles), with the following parameters: 400 million reads depth, 50 Gbases, and 70 million reads per sample.

Cutadapt 1.18 was used to trim nextera adapters in 3' on reads, then a quality control of sequences was done with FastQC. Cellular and UMIs barcodes were extracted with the ddSeeker (v 0.9.0) tool with default parameters. The following steps were done with Drop-seq tool (v 2.0.0). Trimming of 5' adapter sequences and of polyA tails was performed. Unaligned BAM was transformed to fastq with the Picard tool, prior to alignment with STAR on mm10 reference genome. Ddseeker bam outputs previously tagged with molecular/cell barcode were merged with aligned BAM files, according to the Drop-seq tool cookbook. Finally, TagReadWithGeneExonFunction was used to annotate each read with the gene it belongs to, and DigitalExpression was used to count gene transcripts in each cell. The output DGE matrix file is a matrix with a row for each gene, and a column for each cell, containing the number of transcripts observed. This output was loaded into the Seurat (v2) R package for further analysis keeping only cells where at least 200 features were detected, and genes detected in at least 5 cells. The final dataset contains 1740 cells and 15,448 genes. To normalize the data, we applied the global-scaling normalization method "LogNormalize" which normalizes the feature expression measurements for each cell by the total expression, multiplies this by a scale factor (10,000 by default), and log-transforms the result. Then, highly variable genes were detected prior to scaling transformation. To cluster cells, we computed a principal components analysis (PCA) on scaled variable genes, as determined above, using Seurat's *RunPCA* function, and visualized it by computing a Uniform Manifold Approximation and Projection (UMAP) using Seurat's *RunUMAP* function on the top 10 PCs. We used *FindClusters* function with a resolution of 0.6 resulting in 8 clusters. TME and malignant cluster cells were identified according to the expression of the GFP transgene. Then, the *FindAllMarkers* function was used to extract the top differentially expressed genes of each cluster, and to annotate them.

## Single-cell RNA-seq and analysis−10X data

Cells suspension of four dissociated GBMs (2 WT+GCV and 2 p16-3MR +GCV) were loaded with the Chromium Next GEM Chip G Single Cell Kit (10X Genomics, #PN-1000120) and a library was generated using Chromium Next GEM Single Cell 3′ Reagent Kits v3.1 (10X Genomics, #20012850). The library was sequenced on an Illumina NovaSeq 6000 instrument using a 100-cycle S2 flow cell in XP mode, with the following parameters: 2050 million reads depth, 200 Gbases per run, and 50,000 reads per cell.

The Cell Ranger Single-cell Software suite (v.3.0.2) was used to process the data. First, a custom reference genome was created with the mkref function to include 3′LTR and 3MR sequences into the mm10 reference genome. Count function was used on each GEM well that was demultiplexed by *mkfastq* to generate gene-cell matrices. Then, filtered_feature_bc_matrix output was loaded into the Seurat Bioconductor package (v.3.2.3) to filter the datasets and identify cell types using R (v.3.6). Genes expressed in at least five cells and cells with at least 200 features were retained for further analysis. To remove likely dead or multiplet cells from downstream analyses, cells were discarded when they had <500 unique molecular identifiers (UMIs), >60,000 UMIs, or expressed over 8% mitochondrial genes.

All samples were merged together for downstream analysis. As no batch effects were observed among the four samples, no integration step was performed. Gene expression matrix was normalized using the negative binomial regression method implemented in the Seurat *SCTransform* function, via the selection of the top 3000 variable genes and regressed out the mitochondrion expression percentage. The final dataset was composed of 20,293 genes and 26,237 cells.

To cluster cells, we computed a principal components analysis (PCA) on scaled variable genes, as determined above, using Seurat's *RunPCA* function, and visualized it by computing a Uniform Manifold Approximation and Projection (UMAP) using Seurat's *RunUMAP* function on the top 30 PCs. We also computed the *k*-nearest neighbor graph on the top 30 PCs, using Seurat's *FindNeighbors* function with default parameters, and in turn, used Seurat's *FindClusters* function with varying resolution values. We chose a final value of 0.5 for the resolution parameter at this stage of clustering. Clusters were assigned preliminary identities based on the expression of combinations of known marker genes for major cell types. TME clusters were identified with the expression of the *Ptprc* (*Cd45*) gene marker. In order to better identify other cell types, TME cells were removed and a second clustering with a resolution of 0.6 was applied.

The *FindMarkers* function with the default parameters (min.-LogFC = 0.25, min.pct = 0.25, test.use = Wilcox) was used to identify differentially expressed genes in different conditions: (i) p16-3MR +GCV vs. WT+GCV in each cluster; (ii) cells from astrocyte and NP clusters with *Cdkn2a* expression ≥4 (307 cells) vs all the other cells (10,280 cells) in WT+GCV GBMs. Functional enrichment analysis was done with clusterProfiler (v3.14.3) Bioconductor package on the differentially deregulated genes with over-representation analysis (enricher function) and with Gene Set Enrichment Analysis (GSEA function[100]) on all the genes. We searched for ligand/receptor interactions between cluster 0 et *Cd45* positive clusters at 0.5 resolution in our single-cell data, using CellPhoneDB (v.2.1.4). Copy number variations (CNVs) were inferred with inferCNV package (v.1.6.0) with the following parameters: "denoise" and a value of 0.1 for "cutoff".

## Signature expression analyses

We analyzed the senescence signature through our tumoral SCT normalized dataset and three datasets corresponding to patient GBMs[23,26,54]. The Neftel et al., dataset was processed via Seurat (v3.2.3), 10X samples were normalized via the SCT method and Smartseq2 samples were retrieved in log2(TPM + 1). Fastq files for the Bhaduri et al., 10X dataset were processed by cellranger, followed by Seurat analysis with the SCT method. Finally, we retrieved the normalized expression matrix of Johnson and colleagues via synapse (https://www.synapse.org/#!Synapse:syn22257780/wiki/604645).

For both murine and patient GBM datasets, we filtered out transcriptomes expressing the *CD45* (*PTPRC*) gene and pediatric GBMs[23] and we calculated a senescence score resulting from the single-sample GSEA[102] using the R package GSVA (v.1.40.1). For these three datasets, we computed the *z*-scores of the resulting enrichment scores and sliced the signature score distribution into deciles to determine the HIGH senescence cells (last decile), the LOW senescence cells (1st decile), and the others with an average senescence potential (MEDIUM).

## Cox regression analysis

Normalized intensities from TCGA microarray data were obtained from cBioPortal (cbioportal.org), filtering for GBM TCGA, Firehose Legacy dataset. First, single-sample GSEA scores were calculated with R package GSVA (v1.32.0) for senescence genes signature and NRF2 targets signature. Secondly, we fitted a Cox proportional hazards regression model with the *coxph* function from the survival R package (v2.44-1.1), with additional covariates such as p16 copy number alteration (CNA) status, and age of patients. Plots were done with *ggforest* R function.

## Statistical analysis

Data are presented as mean with standard error to the mean (SD) unless otherwise specified. Statistical comparisons were performed using Wilcoxon−Mann−Whitney, *p*-values unless otherwise specified (*$p < 0.05$; **$p < 0.01$; ***$p < 0.001$, ***$p < 0.001$). For Kaplan−Meier

survival curves, statistical significance was determined by the Mantel−Cox log-rank test (*$p < 0.05$). Comparisons between conditions were performed using log-rank tests with a Benjamini−Hochberg correction for multiple comparisons.

### Reporting summary

Further information on research design is available in the Nature Portfolio Reporting Summary linked to this article.

## Data availability

The mm10 reference genome was retrieved from https://hgdownload. cse.ucsc.edu/goldenpath/mm10/chromosomes/. Gene Set Enrichment analysis gene sets came from MSigDB collections (https://software. broadinstitute.org/cancer/software/gsea/wiki/index.php/MSigDB_ collections). The Neftel et al., the dataset was retrieved via the single cell portal (singlecell.broadinstitute.org), fastq files for the Bhaduri et al., dataset, were retrieved from SRA bioproject PRJNA579593, and the normalized expression matrix of Johnson and colleagues was retrieved via synapse (https://www.synapse.org/#!Synapse:syn22257780/wiki/ 604645). Normalized intensities from TCGA microarray data were obtained from cBioPortal (cbioportal.org), filtering for GBM TCGA, Firehose Legacy dataset. The raw data generated in this study are provided in a Source data file and a Source supplementary data file. The data generated in this study have been deposited in the Gene Expression Omnibus database under the accession code GSE168040. All data are available in the article, Supplementary Information and Source Data. Further information and material requests should be addressed to Isabelle Le Roux (isabelle.leroux@icm-institute.org). Source data are provided with this paper.

## Code availability

Computer codes used for the analysis of the senescence score are available following https://github.com/bellenger-l/glioblastoma_ senescence/; https://doi.org/10.5281/zenodo.7525302.

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

## Acknowledgements

The authors acknowledge the service of the ICM platforms: iGenSeq, iVector especially Phillippe Ravassard, icm-QUAN-Imaging, Pheno-ICMice, Histomics-Histology, Celis-Cell culture, the Data Analysis Core platform for their expertise in statistics, in particular, François-Xavier Lejeune and the service of the animal facility UMS28 (Sorbonne University) and the service of the flow cytometry platform CYTO-ICAN especially Florence Deknuydt (Hôpital Pitié-Salpêtrière). The authors are grateful to Judith Campisi, Inder Verma, Magdalena Götz, Anton Berns, and the Centro Nacional de Investigaciones Oncológicas (CNIO) for providing reagents and Carol Schuurmans for comments on the manuscript. We also thank members of the Huillard-Sanson laboratory for discussions, and help with the experiments, in particular Sophie Paris for teaching the intracranial injection. This work was supported by institutional fundings from Paris Brain Institute (ICM), the Institut National de la Santé Et de la Recherche Médicale and the Centre National de la Recherche Scientifique and grants from the ATIP-AVENIR (E.H.), the Cancéropole Ile de France (Emergence Program, I.L.R.), the Ligue contre le cancer, comité Ile de France (I.L.R.); the Fondation ARC pour la Recherche sur le Cancer (E.H., I.L.R.); the SIRIC-CURAMUS (joint Emergence program I.L.R. and C.A.). R. Salam was supported by fellowships from the French Ministry of Education and Research and the Ligue Nationale Contre le Cancer; A. Saliou was supported by a fellowship from the Ligue Nationale Contre le Cancer.

## Author contributions

R.S.: Conceptualization, methodology, experiments, formal analysis, methodology, writing—original draft. A.S.: Conceptualization, methodology, experiments, formal analysis, methodology, writing—original draft. F.B.: Methodology, formal analysis, resources. M.B.: Formal analysis, bioinformatics. C.A.: Formal analysis, bioinformatics, funding acquisition, writing—original draft. C.C.: Resources. A.A.: Bioinformatics. L.C.: Resources. M.S.: Funding acquisition, resources. E.H.: Funding acquisition, resources. L.B.: Formal analysis, bioinformatics, writing—original draft. J.G.: Formal analysis, bioinformatics, writing—original draft. I.L.R.: Conceptualization, methodology, experiments, formal analysis, methodology, funding acquisition, supervision,

## Competing interests

F.B. reports employment of next-of-kin from Bristol-Myers Squibb; research grants from Sanofi and Abbvie outside the submitted work; travel, and accommodations expenses from Bristol-Myers Squibb for travel expenses, outside the submitted work. The remaining authors declare no competing interests.
