## [Peer Review File · Nature Communications]

Cellular senescence in malignant cells promotes tumor progression in mouse and patient GlioblastomaREVIEWER COMMENTS

Reviewer #1 (Remarks to the Author):

This manuscript by Salam et al describes experiments in a mouse genetic model and human gliomas where they have identified a cell population that is high in cdkn2a expression that they then define as senescent. They then kill some of these cells in the mouse model with a transgene expressing TK from the cdkn2a promoter as a means of partly depleting these cells (unknown specificity) and find that the mice live longer. They interpret the results as that senescent cells contribute to glioma aggressiveness. They follow this work by identifying NRF2 as a transcription factor that plays a significant role in the cdkn2a high cells. they conclude by suggesting that drugs targeting senescent cells specifically is a valid therapeutic approach for this tumor type.

The paper is well written, though the observations are a bit counter intuitive. Moreover, there seems to be a bit of a reach in rationale for some of the fundamental claims made. For example:

Senescent cells are defined as not going back into cell cycle, and are high in cdkn2a expression. but there isn't any proof in the experimental model that cells identified by their cdkn2a construct don't go back into cell cycle. With all of the feedback that occurs in the cell cycle components, how certain are the authors that the cdkn2a promoter strength over a particular threshold at any timepoint defines cells specifically as senescent?

And given that there are likely some uncertainties in how specific the promoter strength is for senescent cells, how certain are the authors that ganciclovir treatment in these mice only kills senescent cells? And if there are other cells that are being killed – maybe more quiescent or stem like – then it does seem to be a stretch to say that the difference in survival is due to the effect on senescent cells, which is the whole thesis of the paper.

The NRF2 data is interesting and might begin to provide some mechanistic insight, NRF2 does have strong effects as a transcription factor. It would be interesting to know what is causing the NRF2 activation. In some cancers (but maybe not gliomas) it is mutations to KEAP1. But in others there is metabolic activation of NRF2 by reactive molecules impairing KEAP1 function. Other genetic changes affecting growth signaling (like Kras activation in PDAC, from Tuveson lab) can activate NRF2. Likely these oncogenic effect may cause metabolic stress (ROS, lipid peroxidation, etc.) that cause NRF2. Does loss of cdkn2a directly active NRF2?

Reviewer #2 (Remarks to the Author):

The authors performed an in-depth analysis of cellular senescence in malignant cells in mouse and patient glioblastoma (GBM). After identifying the senescent cells, they remove based on the p16-3MR transgene p16Ink4a-expressing malignant cells in the mouse model and identify a slightly improved median survival. Treatment with ABT263 significantly improved survival in the mouse model. They then performed a detailed characterization of the senescent cells in the mouse model system using bulk and single cell sequencing as well as IHC staining on tumors of the mouse GBM model system. Overall the cellular senescence is described in large detail but the effects on the survival in the mouse model system are only temporarily. Furthermore, the value of their "sen-score" signature could be analysed in more detail as the current multivariate Cox hazard model could be improved.

Major:

1) Fig 2c

The authors performed multiple survival analyses without correcting for multiple testing and specifying the exact test performed. Assuming a log rank test for each of the comparisons a multiple testing correction would be helpful.

2) Fig 7e/g

Did the authors include in their multivariate analysis all variables in addition to their "sen-score" that showed significance in a univariate analysis?

Minor:

A) The authors did not comment on testing the Cox proportional hazards assumption.

Reviewer #3 (Remarks to the Author):

The process of cell senescence has been demonstrated to have context-dependent beneficial or detrimental impact on cancer progression or treatment. In this study the authors investigate the relationship between the presence of TME senescent cells and GBM progression. The authors find that senescent cells contribute to the cellular heterogeneity in GBM. Using strategies to genetically ablate senescent cells the authors find that senescence promote GBM progression. The also show that anti-apoptotic BCL2-family inhibitors (potential senolytic drug) provide a similar effect.

This is an important study elegantly demonstrating the concept that the presence of senescent cells can drive GBM aggressiveness. This study tackle a lethal and difficult to study cancer, GBM, and proposes a new potential therapeutic strategy for this cancer.

The data is dense but relatively well presented and the conclusions appear mostly supported. There are a few caveats that prevent the conclusions from being fully supported and overall a few important prior concepts are evacuated from the paper. For example, it is not novel that "senolytic" or rather BCL2-family inhibitors are effective against GBM. For example these 3 studies rapidly found using a google search (Karpel-Massler - Induction of synthetic lethality in IDH1-mutated gliomas through inhibition of Bcl-xL. *Nat Com* 2017;8(1):1067 ; Tagscherer KE - p53-dependent regulation of Mcl-1 contributes to synergistic cell death by ionizing radiation and the Bcl-2/Bcl-XL inhibitor ABT-737. *Apoptosis*. 2012;17(2):187-99 ; Karpel-Massler G - Inhibition of Mitochondrial Matrix Chaperones and Antiapoptotic Bcl-2 Family Proteins Empower Antitumor Therapeutic Responses. *Cancer Res*. 2017;77(13):3513-26.). Thus the proposed potential "senolytic drug" strategy is not so novel in GBM. This should be discussed and put in the context of the novel discovery here, that maybe these drugs would act via targeting of senescent cells rather than any other GBM cells.

Similarly, there are a few key studies that have investigated one-two punch senolytic strategies against cancer cells. The primary studies, not reviews or older papers using senolytics against age-associated diseases, should be cited. And again put into context with the novelty here (targeting naturally occurring senescent cancer cells versus those created by a prior existing cancer therapies in a one-two punch approach). What are the advantages of the newly proposed approach?

Specific points to strengthen key experiments.

1. In the text, the authors say: "we identified SA- β -gal+ Ki67- LAMINB1- p19ARF+ senescent cells in mouse GBMs (Fig. 1d)." How do you test 4 markers in the same cell? This is not shown... Please adjust and see below for more examples of this weakness.

2. Related point about senescent cell identification. Overall, one key component of this study is the identification of senescent cells in mouse and human GBM. However, none of the presented data convincingly and systematically identify senescent cells (as noted above the confusing reference to quadruple positive cells appear wrong, at best he presented data shows 2 simultaneous markers. Yet, even with 2 markers, there are a lot of representative images where the potential senescent cells are unlabelled and quite difficult or impossible to identify for the reader. These cells should be systematically and properly labelled/identified in the representative images and quantification of the

data presented. For example, are there SABGAL+ Ki67+ cells, if so why? Are they more abundant than the SABGAL+ Ki67- cells? More specific examples: Fig 1 (and s1). It is unclear how senescent cells are quantified, by eye, using the presented images, it is impossible to validate the claims made in the text. For example, in S1b everything looks blue... Please present the quantification of senescent cells or cells positive for the proposed markers. Please describe in more details how the SABGAL staining on fresh tissue slices was followed on the same tissue section by IHC. This is notably difficult to perform as the SABGAL reaction can destroy antigens/epitopes later required during the IHC staining.

3. Related - FigS1D, there are SABGAL+ cells in the full necrotic areas containing mostly dead cells? Isn't this a non-specific staining? There is only one senescent cell in the proliferating area? What point is made? Where are the quantifications matching these representative images?

4. In general, regarding gene expression analysis. Key gene expression data should be validated using qPCR normalized using housekeeping genes known to be stable during senescence (not just RNAseq FKPM estimates or single cell seq). A basic senescence hallmark panel should be validated including p16 and 3MR (the whole strategy is based on p16+ cells elimination this should also be validated in ABT263 treated animals). This was done in fig. 1c it should be done throughout.

5. In the text, the authors say "the difference of survival following a senolytic treatment appeared to be relatively modest, nonetheless this difference was significant and was observed in two replicates using the p16-3MR paradigm and in the ABT263 paradigm. This result is remarkable given the fact that senescent cells represented less than 7% of the tumor and that their removal using the p16-3MR transgene was only partial. These findings suggest that senolytic drug therapy may be a beneficial adjuvant therapy for patients with GBM." This clearance experiment is absolutely key to the paper. The results are promising. Although statistically significant as presented, the differences in survival remain relatively small. It is unclear whether that key experiment was repeated but based on the text above it looks like it was not? If not, it should be repeated and the data presented. The WT+GCV group used to show that GCV alone has no effect on the GBM model used is an excellent and essential control.

Point to point response to the reviewers

Please find the revised version of our manuscript untitled 'Cellular senescence in malignant cells promotes tumor progression in mouse and patient Glioblastoma'. We reached the end of the revision and we are thankful to the reviewers for their useful and constructive comments. We made substantial experimental efforts and editing changes to respond to the points they rose. We do believe that the quality and the clarity of the manuscript greatly improved. Please note that all changes in the revised manuscript text file appear in blue and that some of these changes are reported below.

Reviewer #1 (Remarks to the Author):

This manuscript by Salam et al describes experiments in a mouse genetic model and human gliomas where they have identified a cell population that is high in *cdkn2a* expression that they then define as senescent. They then kill some of these cells in the mouse model with a transgene expressing TK from the *cdkn2a* promoter as a means of partly depleting these cells (unknown specificity) and find that the mice live longer. They interpret the results as that senescent cells contribute to glioma aggressiveness. They follow this work by identifying NRF2 as a transcription factor that plays a significant role in the *cdkn2a* high cells. they conclude by suggesting that drugs targeting senescent cells specifically is a valid therapeutic approach for this tumor type.

The paper is well written, though the observations are a bit counter intuitive. Moreover, there seems to be a bit of a reach in rationale for some of the fundamental claims made. For example:

Senescent cells are defined as not going back into cell cycle, and are high in *cdkn2a* expression. but there isn't any proof in the experimental model that cells identified by their *cdkn2a* construct don't go back into cell cycle. With all of the feedback that occurs in the cell cycle components, how certain are the authors that the *cdkn2a* promoter strength over a particular threshold at any timepoint defines cells specifically as senescent?

We do thank the reviewer for questioning the *Cdkn2a* construct (p16-3MR transgene) used for eliminating $p16^{Ink4a+}$ cells with GCV. We acknowledge that cellular senescence depends on the cell type, the cellular context and no study was performed on GBM with this genetic tool before our work therefore, some clarifications were required.

-In the revised manuscript we now provide evidence that *p16-3MR* expression levels follow the same trend than the endogenous *p16^{Ink4a}* expression levels.

The levels of expression of this transgene were too low to be detected in our single cell 3' RNAseq and bulk RNAseq experiments. We also tried to detect the transgene using the RNAscope technology (ACD Biotechne) on GBM sections but again the low levels of the transcripts prevented precise and efficient segregation of the signal from the background. We thus performed RTqPCR experiments using primers specific to the *p16-3MR* transgene and compared the fold change of the transgene and of *p16^{Ink4a}* between paradigms. We found as written in page 6 that '*Levels of p16-3MR transgene were elevated in the tumor (GFP+) cells compared with the surrounding parenchyma (GFP-)*

cells, similarly to $p16^{\text{Ink4a}}$ expression suggesting that in our model $p16\text{-3MR}$ expression followed the same trend than $p16^{\text{Ink4a}}$ expression (Supplementary Fig.2a)'.

-In the revised manuscript we also clarified the definition of $p16^{\text{Ink4a Hi}}$ cells.

We wrote page 8 '*The $p16\text{-3MR}$ mice were used in different cellular contexts. Injection of GCV always decreased significantly $p16^{\text{Ink4a}}$ levels. However, this decrease never exceeded $p16^{\text{Ink4a}}$ basal expression levels corresponding to the levels observed in the organ in absence of senescent cells^{17,18}. We hypothesized that cells expressing $p16^{\text{Ink4a}}$ at a level ≥ 4 (hereafter, we refer to $p16^{\text{Ink4a Hi}}$ cells) were the cells targeted by $p16\text{-3MR}$ with GCV (Fig. 3d). This threshold was chosen as $p16^{\text{Ink4a Hi}}$ cells represent 3% (412/13563) of the tumor cells, a percentage that is in agreement with the area of SA- β -Gal staining in the tumors (Supplementary Fig. 1a and Fig. 2f)*'.

We further validated our hypothesis by showing in Fig.3 and Supplemental Fig.3 that (i) $p16^{\text{Ink4a Hi}}$ cells were mainly grouped in the astrocyte cluster and that the levels of $p16^{\text{Ink4a}}$ decreased specifically in the astrocyte cluster in $p16\text{-3MR}+\text{GCV}$ GBMs compared with $\text{WT}+\text{GCV}$ GBMs; (ii) the astrocyte cluster shared an inflammatory signature in agreement with a senescent phenotype; (iii) the percentage of cells in the astrocyte cluster decreased in $p16\text{-3MR}+\text{GCV}$ GBMs compared with $\text{WT}+\text{GCV}$ GBMs (astrocyte cluster from 7.75% to 3.21%).

Finally, in Fig. 7, we compared the transcriptome of $p16^{\text{Ink4a Hi}}$ cells with the remaining malignant cells in $\text{WT}+\text{GCV}$ GBMs and extracted a signature composed of genes related to senescence.

-In this study we compared the transcriptome of $p16\text{-3MR}+\text{GCV}$ GBMs with $\text{WT}+\text{GCV}$ GBMs by performing scRNAseq and bulk RNAseq at early timepoint (after one cycle of GCV) and bulk RNAseq at late timepoint (after two cycles of GCV) of tumorigenesis. The main consequence of the senolytic treatment (decreased SASP; plasticity of malignant cells; modulation of the microenvironment; decreased NRF2 signals) were systematically validated by the three transcriptomic analyses. Therefore, we are highly confident that the $p16^{\text{Ink4a Hi}}$ cells are senescent cells at the time we applied the cycles of senolytic treatment.

-As suggested by reviewer#1, some senescent cells may re-enter the cell cycle during the tumor progression. We cannot formally exclude that possibility and only a lineage tracing of these specific cells would give the answer. To date we do not have the right genetic tools to perform such interesting (and long) experiments. Nevertheless, our data strongly suggest that if this process occurs it should concern very few senescent cells. Indeed, the senescent $p16^{\text{Ink4a Hi}}$ signature contains 3 cell cycle inhibitors, namely Cdkn2a, Cdkn2b, Cdkn1a that warrants exit from the cell cycle (Fig. 7c and page 14). In addition, in the revised manuscript we now provide the quantification of SA- β -gal+ cells that do not express the cell cycle marker Ki67 (pages 5-6) and we showed that over 94% of SA- β -gal+ are negative for Ki67 in mouse GBMs and in patient resected tissues.

And given that there are likely some uncertainties in how specific the promoter strength is for senescent cells, how certain are the authors that ganciclovir treatment in these mice only kills senescent cells? And if there are other cells that are being killed – maybe more quiescent or

stem like – then it does seem to be a stretch to say that the difference in survival is due to the effect on senescent cells, which is the whole thesis of the paper.

-We thank reviewer#1 for asking about the cell specificity of the TK killing with GCV. Our scRNAseq analysis provides strong evidence that GCV treatment kills specifically cells in the astrocyte cluster containing cells harboring senescence hallmarks. We wrote page 9: ‘The levels of $p16^{Ink4a}$ decreased significantly in the astrocyte cluster in p16-3MR+GCV GBMs compared with WT+GCV GBMs. No other *malignant and microenvironment (CD45+)* clusters showed a significant difference in $p16^{Ink4a}$ levels between the two conditions (Fig. 3h; *Supplementary Table 2*)’. In the revised manuscript we provide the quantification of $p16^{Ink4a}$ fold change in CD45+ clusters that was missing in the original manuscript (Supplementary Table 2). This result showed that CD45+ cells are very unlikely targeted by the p16-3MR transgene with GCV.

-Thanks to reviewer#1 comments we performed a novel analysis on the bulk RNAseq data to decipher whether senolytic treatment modulate the stemness signature. GSEA analysis on bulk RNAseq (early and late timepoints) using the stemness gene signature from Tirosh *et al.*, (2016; *Supplementary Table 1*) showed no enrichment in p16-3MR+GCV GBMs compared with WT+GCV controls. This result strongly suggests that p16-3MR+GCV does not target glioma stem cells. We now included this analysis to the GSEA ridge plot shown in Fig.3j.

The NRF2 data is interesting and might begin to provide some mechanistic insight, NRF2 does have strong effects as a transcription factor. It would be interesting to know what is causing the NRF2 activation. In some cancers (but maybe not gliomas) it is mutations to KEAP1. But in others there is metabolic activation of NRF2 by reactive molecules impairing KEAP1 function. Other genetic changes affecting growth signaling (like Kras activation in PDAC, from Tuveson lab) can activate NRF2. Likely these oncogenic effect may cause metabolic stress (ROS, lipid peroxidation, etc.) that cause NRF2. Does loss of *cdkn2a* directly active NRF2?

We acknowledge reviewer#1 for asking more details about NRF2 activation in the context of GBM. In the revised manuscript we added new information in the discussion section. We wrote page 16: ‘*Impairment of NRF2-KEAP1 binding, either by phosphorylated-p62/SQSTM1 or by elevated ROS permits NRF2 nuclear translocation and subsequent activation of target genes^{55,56,57}. Of note, p62/SQSTM1-mediated degradation of KEAP1 and NRF2 promotes in vitro glioma stem cell survival⁵⁸. NRF2 is hyperactivated preferentially in the MES-GBM subtype⁵⁸ however, NRF2/NFE2L2 and genes regulating its activity (KEAP1, SQSTM1, CUL3, RBX1, SKP1, CUL1, BTRC, SYVN1) are rarely altered in GBMs (see cbioportal.org). The putative triggers of senescence in our GBM model and in patient GBMs are also known to regulate NRF2 activity such as hypoxia, ROS, PI3K-AKT pathway (enhanced by the loss of PTEN)^{48,59} or Nrf2 transcription such as RAS oncogene (K-RAS)⁶⁰. Future work is required to establish the contribution of these triggers in the process of cellular senescence and NRF2 activity in GBMs*’.

We showed in the manuscript (Fig. 5) that the *Nfe2l2/NRF2* signaling pathway was enhanced in the three gene sets enriched in p16^{Ink4a}^{HI} malignant cells and the Cox regression model showed (Fig.7g) that regardless of p16^{Ink4a} status, an enriched NRF2 target gene score predicted a worse survival in patients with GBM. Therefore, it was unlikely that loss of *Cdkn2a* directly activates NRF2. Indeed, our novel analysis from TCGA GBM data sets (microarrays data; n=508 patients) showed an absence of correlation between the copy number alteration of *Cdkn2a/p16^{Ink4A}* (cna; X axis; see the figure below) and the *NRF2* expression (Y axis; see the figure below). Statistical significance was determined by Wilcoxon-Mann-Whitney test. We did not include the data in the result section but this can be done.

Reviewer #2 (Remarks to the Author):

The authors performed an in-depth analysis of cellular senescence in malignant cells in mouse and patient glioblastoma (GBM). After identifying the senescent cells, they remove based on the p16-3MR transgene p16Ink4a-expressing malignant cells in the mouse model and identify a slightly improved median survival. Treatment with ABT263 significantly improved survival in the mouse model. They then performed a detailed characterization of the senescent cells in the mouse model system using bulk and single cell sequencing as well as IHC staining on tumors of the mouse GBM model system.

Overall the cellular senescence is described in large detail but the effects on the survival in the mouse model system are only temporarily. Furthermore, the value of their “sen-score” signature could be analysed in more detail as the current multivariate Cox hazard model could be improved.

We do thank the reviewer for the careful statistical review. We added all their pertinent suggestions to the revised manuscript.

Major:

1) Fig 2c

The authors performed multiple survival analyses without correcting for multiple testing and specifying the exact test performed. Assuming a log rank test for each of the comparisons a multiple testing correction would be helpful.

We agree with the reviewer's remark and we addressed this issue. Statistical significance was determined by Mantel-Cox log-rank test as written in the figure legend page 33. We confirmed that all the three significant differences a/a', b/b' and a'/b mentioned in Fig.2c remained valid after a Benjamini-Hochberg correction to control for the false discovery rate. While we still indicate the raw p-values from the log-rank tests on the figure, we corrected one significance level from ** to * as the three BH-corrected p-values were < 0.05. The explanation was also added to the legend page 33.

For information, the exact values are:

Raw p-values: 0.005, 0.017, 0.014, 0.072, 0.448, 0.524

BH-adjusted p-values (same order): 0.030, 0.034, 0.034, 0.108, 0.524, 0.524

2) Fig 7e/g

Did the authors include in their multivariate analysis all variables in addition to their “sen-score” that showed significance in a univariate analysis?

In the revised manuscript we now include the Karnofsky score and the Sex as variables for the multivariate Cox regression model as these variables are commonly used for patients with GBM. We generated a new Supplementary table (Supplementary Table 4) related to these analyses and show the univariate analysis of all the variables showed in Fig. 7e, g.

Minor:

A) The authors did not comment on testing the Cox proportional hazards assumption.

The Schoenfeld residual test using the R function 'cox.zph' did not indicate significant departure with Cox proportional hazard assumptions, except for the Age covariate (p=0.03). We now include these results in the new Supplementary Table 4. To put this difference in perspective, Age is not the main tested effect, and was introduced for covariate adjustment, with p16 copy number alteration, Sex and Karnofsky score. We can see graphically (see below, the Kaplan-Meier survival curves) that the difference between the survival curves of the two age groups (age<=59 and age>59) remains relatively constant before joining at the very end of the observation period (after 100 days) at the 0 percent level.

Reviewer #3 (Remarks to the Author):

The process of cell senescence has been demonstrated to have context-dependent beneficial or detrimental impact on cancer progression or treatment. In this study the authors investigate the relationship between the presence of TME senescent cells and GBM progression. The authors find that senescent cells contribute to the cellular heterogeneity in GBM. Using strategies to genetically ablate senescent cells the authors find that senescence promote GBM progression. The also show that anti-apoptotic BCL2-family inhibitors (potential senolytic drug) provide a similar effect.

This is an important study elegantly demonstrating the concept that the presence of senescent cells can drive GBM aggressiveness. This study tackle a lethal and difficult to study cancer, GBM, and proposes a new potential therapeutic strategy for this cancer.

The data is dense but relatively well presented and the conclusions appear mostly supported. There are a few caveats that prevent the conclusions from being fully supported and overall a few important prior concepts are evacuated from the paper. For example, it is not novel that “senolytic” or rather BCL2-family inhibitors are effective against GBM. For example these 3 studies rapidly found using a google search (Karpel-Massler - Induction of synthetic lethality in IDH1-mutated gliomas through inhibition of Bcl-xL. *Nat Com* 2017;8(1):1067 ; Tagscherer KE - p53-dependent regulation of Mcl-1 contributes to synergistic cell death by ionizing radiation and the Bcl-2/Bcl-XL inhibitor ABT-737. *Apoptosis*. 2012;17(2):187-99 ; Karpel-Massler G - Inhibition of Mitochondrial Matrix Chaperones and Antiapoptotic Bcl-2 Family Proteins Empower Antitumor Therapeutic Responses. *Cancer Res*. 2017;77(13):3513-26.). Thus the proposed potential “senolytic drug” strategy is not so novel in GBM. This should be discussed and put in the context of the novel discovery here, that maybe these drugs would act via targeting of senescent cells rather than any other GBM cells.

We do thank reviewer#3 for suggesting insightful discussion elements on the use of anti-apoptotic drugs in the context of GBM *in vivo*. We substantially remodeled the discussion section accordingly. We now include two new paragraphs in the discussion on these findings, we wrote page 17-18:

'In this study we showed that senolytic treatments applied to GBM bearing mice delays temporarily tumor growth (Fig. 2c and d). Previous work investigated the action of the specific inhibitors of the anti-apoptotic BCL2 and BCL-xL proteins, ABT737 and ABT263, in the context of GBM mouse models. Similarly to our results (Fig. 2d), ABT737 treatment increases the survival of immunodeficient mice grafted with the human glioma cell line U-251MG⁷⁰. ABT263 treatment when combined with drugs decreasing Mcl1 levels (a BCL2 family protein), decreased tumor volume in a heterotopic (subcutaneous) model of proneural GBM^{71,72}. Furthermore, ABT263 treatment provides a survival benefit only when applied in combination with the oncometabolite 2-R-2-hydroxyglutarate (2-HG; produced by IDH1-mutated tumors) in a proneural GBM model⁷³. Cellular senescence was not addressed in these studies but in the light of our results using the p16-3MR transgene, some of the cells targeted by this approach may be malignant senescent cells, extending the notion of detrimental senescent cells to distinct GBM subtypes in agreement with the senescence score analysis performed on data from patient GBMs (Figure 7). All together our findings and these previous studies, raise the possibility to use senotherapy to improve the outcome of patient with GBM'.

'Many important issues remain to be solved before envision senotherapy in the context of GBM such as the nature of the senolytic, the timing of its administration (neoadjuvant-concomitant adjuvant or adjuvant) and the most effective combination of treatment. First of all, these drugs should cross the brain-blood-barrier. Novel molecules are expected to be discovered in the near future as the field of senotherapies, which includes drugs eliminating senescent cells (senolytic drugs: anti- BCL2 and BCL-xL¹³, dasatinib and quercetin¹⁴, cardiac glycosides^{74,75,76}, CAR T cells⁷⁷) and drugs inhibiting their function (senostatic drugs such as Metformin⁷⁸) is under active investigation⁷⁹. Despite encouraging results in mouse GBM models, clinical studies show major side-effects of ABT263 such as thrombocytopenia and neutropenia caused by BCL-xL inhibition⁸⁰, limiting the use of BCL-xL inhibitors as safe and effective anticancer agents. The selective BCL2 inhibitor ABT199 (Venetoclax) which spares the platelets, displays variable results on senescent glioma cells, in vitro^{81,82}. Administration of a combination of dasatinib and quercetin, a multi-tyrosine kinase inhibitor and a natural flavonoid respectively, eliminates efficiently senescent cells in different pathologies in the mouse including neurodegenerative diseases^{83,84}. Results of ongoing clinical trials using these drugs on cohorts of patients with mild cognitive impairment and Alzheimer disease (NCT04785300i, NCT04685590ii) should encourage or refute this approach for targeting senescence in the central nervous system'.

Similarly, there are a few key studies that have investigated one-two punch senolytic strategies against cancer cells. The primary studies, not reviews or older papers using senolytics against age-associated diseases, should be cited. And again put into context with the novelty here (targeting naturally occurring senescent cancer cells versus those created by a prior existing cancer therapies in a one-two punch approach). What are the advantages of the newly proposed approach?

Again, we acknowledge reviewer#3 for suggesting insightful discussion element. We now include a new paragraph on the one-two punch senolytic strategies against cancer

cells and discuss our findings and the nature of glioblastoma in the light of these experiments. We wrote page 18:

'Heterotopic and orthotopic mouse models of cancers showed the efficacy of the one two punch sequential therapy defined by a pro-senescence therapy, followed by senolytic therapy clearing the induced-senescent cells and preventing the accumulation of detrimental persistent senescent cells^{85,18,86,77,87}. The efficacy of such strategy implies a homogeneous response to the pro-senescence therapy. GBMs which are characterized by intra-tumoral heterogeneity are thus not the ideal candidate for the use of this strategy. Nonetheless, radiation and TMZ chemotherapy induce senescence in glioma cells in vitro and sensitize these cells to anti-apoptotic inhibitors with distinct efficacy according to the patient-derived cell lines, suggesting that this strategy may be relevant for some patients with GBM^{88,82}. Based on our work which focused on naturally-occurring senescence, we hypothesize that senolytics combined to conventional treatment could have a double action by depleting both resident and therapy-induced senescent cells which may potentialize their effect. Also, we would like to propose another combined therapeutical strategy that would take advantage of the tumor ecosystem modifications following senolytic treatment. For example, promyelinating drugs may amplify the plasticity of malignant cells towards an oligodendroglial-like differentiated phenotype initiated by the senolytic treatment (Figure 3). Another strategy could be to target the immune cells. Indeed, although a thorough study on the consequence of senolytic treatment on the immune system is required, our study provides evidence of a decreased anti-inflammatory phenotype after treatment (Figure 4). Therefore, senolytic treatment may prime GBM to respond to immunotherapy. This hypothesis is attractive as the immunotherapies with the anti-PD1 and PD-L1 antibodies did not show an extension of the overall survival in treating patients with recurrent GBM^{89,90}. One possible explanation for this failure could be that GBMs contain very few immune effector cells⁹¹. Further work on immunocompetent GBM models, reproducing the intra-heterogeneity of patient GBMs, is now needed to evaluate the effect of novel senotherapy on glioma progression and assess their efficacy as companion therapy.'

Specific points to strengthen key experiments.

1. In the text, the authors say: “we identified SA-β-gal+ Ki67- LAMINB1- p19ARF+ senescent cells in mouse GBMs (Fig. 1d).” How do you test 4 markers in the same cell? This is not shown... Please adjust and see below for more examples of this weakness.

2. Related point about senescent cell identification. Overall, one key component of this study is the identification of senescent cells in mouse and human GBM. However, none of the presented data convincingly and systematically identify senescent cells (as noted above the confusing reference to quadruple positive cells appear wrong, at best he presented data shows 2 simultaneous markers. Yet, even with 2 markers, there are a lot of representative images where the potential senescent cells are unlabelled and quite difficult or impossible to identify for the reader. These cells should be systematically and properly labelled/identified in the

representative images and quantification of the data presented. For example, are there SABGAL+ Ki67+ cells, if so why? Are they more abundant than the SABGAL+ Ki67- cells?

More specific examples: Fig 1 (and s1). It is unclear how senescent cell are quantified, by eye, using the presented images, it is impossible to validate the claims made in the text. For example, in S1b everything looks blue... Please present the quantification of senescent cells or cells positive for the proposed markers.

We agree with the reviewer's remarks, we did not perform quadruple staining on the same section; we adjusted the main text file accordingly. The major flaw of the SA- β -gal staining is that it is chromogenic, preventing multiple co-stainings. Thus, we tried to label senescent cells using a fluorescent probe (CellEvent Senescence Green Detection kit #C10850, ThermoFischer) in order to co-stain cells with different fluorescent markers. However, after several optimizations on mouse and patient glioblastoma sections, we were not convinced by the specificity of the fluorescent senescent probe and did not follow up with this experiment.

In the revised manuscript we improve the identification of senescent cells by quantifying as requested by reviewer#3, the number of SA- β -gal + cells positive for the cell cycle marker Ki67 and the cell cycle inhibitors p16^{Ink4a} (for patient samples) and p19^{Arf} (for mouse GBMs). We focused on these markers as they are essential to define cellular senescence. Of note, we did not quantify the number of SA- β -gal + cells positive for GFAP and IBA1, as the membrane staining of these proposed markers did not allow an accurate quantification of individual cells. We found that above 94% of SA- β -gal + cells were negative for Ki67 and more than 72% of SA- β -gal + cells were positive for the cell cycle inhibitors in patient samples and mouse GBMs. Therefore these *'results strongly suggest that the majority of SA- β -gal+ cells were senescent'*. We included these quantifications in the main text file (page 5-6).

To establish a quantitative measure of senescent cell burden, we quantified the percentage of the tumor area containing SA- β -gal+ cells. This method is commonly used to quantify senescence as this staining is mostly cytoplasmic (see for example Munoz-Espin et al., 2013 doi: 10.1016/j.cell.2013.10.019). We acknowledge that senescence is not defined by one marker but by a combination of markers. The quantifications of cells SA- β -gal +Ki67- and SA- β -gal +p16^{Ink4a}/p19^{Arf}+ strongly suggest as mentioned above, that SA- β -gal staining in the context of GBM is a good marker of senescence. The detailed surface area quantification of SA- β -gal using Fiji software is provided page 22 in the Method section. Importantly, *'IHC images were color deconvoluted according to the "Giemsa" or "Hematoxylin and DAB (H DAB)" vector to assess a threshold of the SA- β -gal or DAB signal respectively. In addition, the signal threshold was adjusted in order to remove the unspecific background signal without clearing the specific one'*. The raw data of the quantifications are available in the source file corresponding to Supplementary Fig.1a and Fig.2e-f.

In Fig. S1b, we acknowledge that the counter staining with Hematoxylin is strong and could have prevented reviewer#3 to clearly distinguish SA- β -gal + from the Hematoxylin

staining. In the revised manuscript, we have now properly labelled the SA- β -gal+ cells in the representative images Fig. 1 a and d and Supplementary Fig.1. b and d.

Please describe in more details how the SABGAL staining on fresh tissue slices was followed on the same tissue section by IHC. This is notably difficult to perform as the SABGAL reaction can destroy antigens/epitopes later required during the IHC staining.

We acknowledge that this important information was missing in the original version of the manuscript. We added in the Method section page 21 : '*Note, that the concentration of glutaraldehyde was dropped from 0.2% to 0.02% from the original protocol⁹⁷ to allow combined immunohistochemical staining*'.

3. Related - FigS1D, there are SABGAL+ cells in the full necrotic areas containing mostly dead cells? Isn't this a non-specific staining? There is only one senescent cell in the proliferating area? What point is made? Where are the quantifications matching these representative images?

The point of Fig.S1d was to illustrate that SA- β -gal+ cells '*were sparsely distributed in the tumor, and mostly located in proliferative areas or adjacent to necrotic regions*' (page 6).

We agree with reviewer#3, the picture in Fig S1d (left panel) was not well chosen to illustrate the presence of SA- β -gal+ cells adjacent to necrotic area. SA- β -gal staining could diffuse even if we performed a post-fixation step (4% PFA 10 min at RT), and this figure illustrated in part, that point. Importantly, for quantification of the SA- β -gal+ area over the tumor area, we adjusted the signal threshold in order to remove the unspecific background signal. In the revised manuscript we included a new picture to better illustrate the presence of SA- β -gal+ cells adjacent to necrotic area. Of note, Fig.6f illustrates also that point.

The picture in Fig.S1d (right panel) illustrated the presence of SA- β -gal+ cells in a proliferative area, labelled with many Ki67+ cells. In the revised manuscript, we labelled the SA- β -gal+ cells in the representative image. Fig.1d (top, left panel) represented a higher magnification of double labelling of SA- β -gal and Ki67 and indeed, this image contained only one SA- β -gal cells. To be more representative we changed this high magnification picture in the revised manuscript. As mentioned above, we included the quantifications of SA- β -gal+ cells negative for Ki67 in the main text file (page 5-6).

4. In general, regarding gene expression analysis. Key gene expression data should be validated using qPCR normalized using housekeeping genes known to be stable during senescence (not just RNAseq FKPM estimates or single cell seq). A basic senescence hallmark panel should be validated including p16 and 3MR (the whole strategy is based on p16+ cells elimination this should also be validated in ABT263 treated animals). This was done in fig. 1c it should be done throughout.

We now provide in the revised manuscript RTqPCR experiments on key gene expression data as requested by reviewer#3. These results further confirmed our scRNAseq and bulk RNAseq analyses. We included these results in the main text file and in the following figures:

-Fig. 2h: relative mRNA levels of cell cycle inhibitors.

-Fig. S2h: relative mRNA levels of SASP.

-Fig. 4d: relative levels of bone marrow-derived macrophages and microglia markers.

In addition, thanks to reviewer#3 comment, we performed RTqPCR experiments using primers specific to the p16-3MR transgene and compared the fold change of the transgene and of $p16^{Ink4a}$ between paradigms. Of note, the levels of expression of this transgene were too low to be detected in our single cell 3' RNAseq and bulk RNAseq experiments. We also tried to detect the transgene using the RNAscope technology (ACD Biotechne) on GBM sections but again the low levels of the transcripts prevented precise and efficient segregation of the signal from the background. However, by performing RTqPCR, we found as written in page 6 that '*Levels of p16-3MR transgene were elevated in the tumor (GFP+) cells compared with the surrounding parenchyma (GFP-) cells, similarly to $p16^{Ink4a}$ expression suggesting that in our model p16-3MR expression followed the same trend than $p16^{Ink4a}$ expression (Supplementary Fig.2a)*'.

To understand and unveil the underlying mechanism of the tumor promoting function of senescent cells, we focused our analyses on p16-3MR+GCV GBMs compared with WT+GCV GBMs for all the experiments and on p16-3MR+GCV GBMs compared with p16-3MR+vhc GBMs for bulk RNAseq at the late timepoint. We did not perform further experiments on the ABT263 treated mice than the survival analysis. Indeed, the ABT263 increased significantly the GBM-bearing mouse survival but it is also known to induce major side effects. Please see also above and page 17-18 our new elements of discussion on the ABT263 paradigm.

5. In the text, the authors say “the difference of survival following a senolytic treatment appeared to be relatively modest, nonetheless this difference was significant and was observed in two replicates using the p16-3MR paradigm and in the ABT263 paradigm. This result is remarkable given the fact that senescent cells represented less than 7% of the tumor and that their removal using the p16-3MR transgene was only partial. These findings suggest that senolytic drug therapy may be a beneficial adjuvant therapy for patients with GBM.” This clearance experiment is absolutely key to the paper. The results are promising. Although statistically significant as presented, the differences in survival remain relatively small. It is unclear whether that key experiment was repeated but based on the text above it looks like it was not? If not, it should be repeated and the data presented. The WT+GCV group used to show that GCV alone has no effect on the GBM model used is an excellent and essential control.

We acknowledge that the sentence quoted by reviewer#3 lacked clarity. We now wrote page 16: '*The difference of survival following a senolytic treatment appeared to be relatively modest, nonetheless this difference was significant and was observed in the p16-3MR paradigm using two distinct controls (WT+GCV and p16-3MR+vhc) and in the ABT263 paradigm*'. All the subsequent analyses presented in the manuscript were

performed on the p16-3MR+GCV group compared the WT+GCV group as controls and bulk RNAseq experiment was also performed at the late timepoint on the p16-3MR+GCV group compared with the p16-3MR+vhc group (Fig. S2d,e,g).

In addition, we now confirm that all the three significant differences a/a', b/b' and a'/b mentioned in the survival study in Fig.2c remained valid after a Benjamini-Hochberg correction to control for the false discovery rate. While we still indicate the raw p-values from the log-rank tests on the figure, we corrected one significance level from ** to * as the three BH-corrected p-values were < 0.05. The explanation was also added to the legend page 33.

REVIEWERS' COMMENTS

Reviewer #1 (Remarks to the Author):

it appears that the authors have addressed the concerns of the reviewers.

Reviewer #2 (Remarks to the Author):

The authors resolved all my open questions and included additional clarifications for the statistical tests performed.

Reviewer #3 (Remarks to the Author):

Dear authors, I have read with attention your response to my previous comments and I find that overall the points that I have raised were appropriately addressed.

A possible exception is point #5, for which I consider that a decision has to be made by the editors based on the standard in their journal regarding the repetition of difficult to perform mouse experiments.

As a note, I am perfectly comfortable with the presented results in figure 2c and 2d, which look strong and support the overall conclusion of the manuscript.

A brief reminder regarding point #5 concerning mice survival experiments: Summary of my previous comment - "This clearance experiment is absolutely key to the paper. The results are promising. Although statistically significant as presented, the differences in survival remain relatively small. It is unclear whether that key experiment was repeated but based on the text above it looks like it was not? If not, it should be repeated and the data presented."

I find the response to comment #5 quite clear, the survival experiment as presented in figure 2c and 2d remain significant after multiple types of analysis suggested by other reviewers. Each figure is presented separately for clarity which make sense, figure 2 c and 2d represent 2 test of different experimental strategies to reach a similar end (one for clearance by ABT and one for clearance by GCV). Although this suggest that 2 different strategies to clear senescent cells reach the same result, it is in no way certain that the ABT experiment clears only senescent cells (unlike the genetic approach that kill p16+ cells). In the end, each experiment was performed only 1 time, with no repetition.

Point to point response to the reviewers

We are grateful to all reviewers for their positive comments on our revised manuscript.

Reviewer #1 (Remarks to the Author):

it appears that the authors have addressed the concerns of the reviewers.

Reviewer #2 (Remarks to the Author):

The authors resolved all my open questions and included additional clarifications for the statistical tests performed.

Reviewer #3 (Remarks to the Author):

Dear authors, I have read with attention your response to my previous comments and I find that overall the points that I have raised were appropriately addressed.

A possible exception is point #5, for which I consider that a decision has to be made by the editors based on the standard in their journal regarding the repetition of difficult to perform mouse experiments.

As a note, I am perfectly comfortable with the presented results in figure 2c and 2d, which look strong and support the overall conclusion of the manuscript.

A brief reminder regarding point #5 concerning mice survival experiments: Summary of my previous comment - "This clearance experiment is absolutely key to the paper. The results are promising. Although statistically significant as presented, the differences in survival remain relatively small. It is unclear whether that key experiment was repeated but based on the text above it looks like it was not? If not, it should be repeated and the data presented."

We apologize if the main text lacked clarity. We now wrote page 16: 'The difference of survival following a senolytic treatment appeared to be relatively modest, nonetheless this difference was significant and was observed in the p16-3MR paradigm when compared to a first control cohort (WT+GCV). These results were repeated using a second control cohort (p16-3MR+vhc). In addition, we observed a benefic effect of another senolytic, ABT-263, in a treated-cohort compared to a control one'.

I find the response to comment #5 quite clear, the survival experiment as presented in figure 2c and 2d remain significant after multiple types of analysis suggested by other reviewers. Each figure is presented separately for clarity which make sense, figure 2 c and 2d represent 2 test of different experimental strategies to reach a similar end (one for clearance by ABT and one for clearance by GCV). Although this suggest that 2 different strategies to clear senescent cells reach the same result, it is in no way certain that the ABT experiment clears only senescent cells (unlike the genetic approach that kill p16+ cells). In the end, each experiment was performed only 1 time, with no repetition.

As mentioned above the p16-3MR paradigm was repeated twice whereas the ABT263 paradigm was done only once on a cohort of 11 treated mice compared with a cohort of 11 control mice. Statistical significance was determined by Mantel-Cox log-rank test and the p value between the ABT263 treated and the control groups was $p=0.022$.